# Description and Evaluation of the JULES-ES setup for ISIMIP2b

Camilla Mathison[1,2], Eleanor Burke[1], Andrew Hartley[1], Douglas I Kelley[4], Chantelle Burton[1], Eddy Robertson[1], Nicola Gedney[1], Karina Williams[1,3], Andy Wiltshire[1,3], Richard J. Ellis[4], Alistair A. Sellar[1], Chris D. Jones [1]

[1] Met Office Hadley Centre, FitzRoy Road, Exeter, UK
[2] School of Earth and Environment, Institute for Climate and Atmospheric Science, University of Leeds, Leeds, UK
[3] Global Systems Institute, University of Exeter, Laver Building, North Park Road, Exeter, UK
[4] UK Centre for Ecology and Hydrology, Wallingford, Oxfordshire, OX10 8BB, UK

*Correspondence to*: Andrew Hartley (andrew.hartley@metoffice.gov.uk)

**Abstract.** Global studies of climate change impacts that use future climate model projections also require projections of land surface changes. Simulated land surface performance in Earth System models is often affected by the atmospheric models' climate biases, leading to errors in land surface projections. Here we run the Joint UK Land Environment Simulator Earth System configuration (JULES-ES) land surface model with Inter Sectoral Impacts Model Intercomparison Project 2[nd] phase future projections (ISIMIP2b) bias-corrected climate model data from 4 global climate models (GCMs). The bias correction

reduces the impact of the climate biases present in individual models. We evaluate JULES-ES performance against present-day observations to demonstrate its usefulness for providing required information for impacts such as fire and river flow. We include a standard JULES-ES configuration without fire as a contribution to ISIMIP2b and JULES-ES with fire as a potential future development. Simulations for gross primary productivity (GPP), evapotranspiration (ET) and albedo compare well against observations. Including fire improves the simulations, especially for ET and albedo and vegetation distribution, with

some degradation in shrub cover and river flow. This configuration represents some of the most current earth system science for land surface modelling. The suite associated with this configuration provides a basis for past and future phases of ISIMIP, providing a simulation setup, postprocessing and initial evaluation using the International Land Model Benchmarking (ILAMB) project. This suite ensures that it is as straightforward, reproducible and transparent as possible to follow the protocols and participate fully in ISIMIP using JULES.

## 1 Introduction

The Joint UK Land Environment Simulator (JULES) (Clark et al., 2011; Best et al., 2011) is a community-supported and developed land surface model used by land, hydrological, weather and climate communities. JULES is a configurable code base supporting weather, climate and earth system science applications. Here, we describe and evaluate the JULES Earth System (JULES-ES) configuration and experimental setup used in the Inter-Sectoral Impact Model Intercomparison Project

(ISIMIP; (Frieler et al., 2017)). JULES-ES builds on the JULES-GL7 configuration described in (Wiltshire et al., 2020) by including additional biogeochemical fluxes governing carbon and nitrogen cycles that influence earth system processes; these

are considered more relevant to ecosystems and people, which could be affected by climate change. Whilst we run JULES-ES in offline mode, it is also coupled to the atmosphere within the earth system model UKESM (Sellar et al., 2019). Climate change impacts are already a feature of everyday life for much of the world, and quantifying these allows us to understand future benefits and trade-offs of climate mitigation and adaptation policies. ISIMIP provides a consistent framework for assessing impacts using a large ensemble of impacts models across various sectors (Warszawski et al., 2013, 2014). ISIMIP has recently completed its second phase, having more than 60 modelling groups contributing simulations to ISIMIP2a (reanalysis-driven hindcasts) and ISIMIP2b (bias-corrected GCM-driven historical and future scenarios). We present the JULES-ES configuration and experimental setup that has contributed to ISIMIP2b and will be the basis for further development in subsequent ISIMIP phases. An advantage of using JULES-ES as an offline impacts model is that it uses a prescribed atmosphere; without these feedbacks JULES-ES is more computationally efficient (10-20k) than the closely aligned land surface scheme coupled to the atmosphere in UKESM1 (Sellar et al., 2019), and using multi-model climate ensembles sample scientific uncertainty in our understanding of the climate system that would not be possible within a single climate model framework.

This paper briefly describes the changes to JULES-GL7 (Wiltshire et al., 2020) that form the JULES-ES configuration, the ISIMIP setup and an evaluation of the arising simulations. JULES-ES has been widely evaluated and applied for global biogeochemical modelling (Sellar et al., 2019; Slevin et al., 2017), including in the Global Carbon Budget (Friedlingstein et al., 2020). Here we focus on using JULES for impacts applications. Alongside this manuscript, we provide a suite to run JULES-ES following the ISIMIP2b modelling protocol (Frieler et al., 2017) for tailored impact projections that are consistent across sectors such as water and biomes. The suite includes the code to post-process output into ISIMIP formatted netcdf output and run the International Land Model Benchmarking (ILAMB) system to allow for quick evaluation (see Text S1). Data from the JULES-ES ISIMIP2b suite have been submitted to the biomes and water ISIMIP2b sectors and are available via the ISIMIP model archive (https://data.isimip.org/) and provide simulations of the historical and future land surface in the (United Nations Environment Programme, 2022) wildfire report. The JULES ISIMIP2b simulations with fire provide the basis for the contribution of JULES to the next FireMIP, which will use the ISIMIP3 set-up. The historical simulations and their evaluation are shown in section 3, with discussion and conclusions in Sections 4 and 5 respectively.

## 2 Materials and Methods

### 2.1 JULES-ES Configuration

To better represent variation in plant traits and managed land, we extend the standard representation of 5 Plant Functional Types (PFTs) to 13 in JULES-ES, building on (Harper et al., 2016), with 4 managed and 9 natural PFTs. Natural PFTs are extended by splitting trees into deciduous and evergreen types and then distinguishing between temperate and tropical

broadleaf evergreen trees. These additional PFTs represent a wider range of leaf life spans and metabolic capacities. Evergreen
trees typically have less access to nutrients, higher leaf mass per unit area, longer lifespans and lower carbon assimilation and
respiration rates, whereas a deciduous PFT typically has leaves with a higher nutrient concentration, shorter lifespan and lower
leaf mass per unit area. Tropical broadleaf evergreen trees have lower maximum carbon assimilation rates than temperate trees.
The 9 natural PFTs used are: tropical broadleaf evergreen trees (BET-Tr), temperate broadleaf evergreen trees (BET-Te),
broadleaf deciduous trees (BDT), needle-leaf evergreen trees (NET), needle-leaf deciduous trees (NDT), C3 grasses (C3G),
C4 grasses (C4G), evergreen shrubs (ESh), and deciduous shrubs (DSh). (Harper et al., 2016) also updated several parameters
required for calculating photosynthesis and respiration using the TRY database (Kattge et al., 2011). They also reduced the
bias in model simulations by tuning parameters relating to leaf dark respiration, canopy radiation, canopy nitrogen, stomatal
conductance, root depth, and temperature sensitivities of the maximum carboxylation rate of Rubisco (Vcmax) based on
available observations. The 4 managed PFTs are C3 and C4 crop and pasture (C3Cr, C4Cr, C3Pa, C4Pa) PFTs and are
functionally similar to natural grasses but, in the case of crops, are assumed not to be nitrogen limited and litter carbon is
removed as a simple representation of crop harvest  (Sellar et al., 2019; Robertson, 2019). This simple fertilization of crops
through lack of nitrogen limitation does not distinction between mineral and manure fertilizer. Both crop and pasture surface
types undergo land-use change according to externally forced time-varying land use, but the pasture is not grazed and is
otherwise unmanaged. Within the respective crop/pasture fraction, only the C3 and C4 crop/pasture PFTs are allowed to grow,
with the area of each determined by the TRIFFID dynamic vegetation module (Burton et al., 2019).

Outside of the managed land area, the nine natural PFTs (including natural C3 grasses and C4 Grasses) can grow in the
remainder of the grid box once the non-vegetated surfaces have been accounted for (urban, ice, lakes, represented by
corresponding non-vegetation surface types, and ocean which is not simulated). As the net prescribed crop or pasture fraction
increases with land-use change, natural vegetation is removed from the portion of the grid box into which agriculture has
expanded, representing anthropogenic land clearance. Conversely, when crop and pasture areas are reduced, the natural PFTs
are allowed to recolonize the vacated grid box fraction. The vacated gridbox fraction is initially bare soil and the existing
natural PFTs gradually expand their coverage into this area; the rate of expansion is determined by the TRIFFID vegetation
dynamics scheme and will follow a succession of faster growing grass PFTs, followed by shrubs and then trees. Bare soil
occupies any remaining space once the vegetation dynamics have been simulated. Simple representations of fertilisation and
harvesting are applied to the crop PFTs, but otherwise these are physiologically identical to the natural grasses. After
accounting for land-use, the fractional coverage and biomass of each PFT within a grid box is determined by the TRIFFID
dynamic vegetation model. Inter-PFT competition is based on vegetation height, with the taller vegetation shading and
therefore dominating other PFTs (Harper et al., 2018).

The other major change introduced in JULES-ES is a representation of nitrogen and nutrient limitation effects on ecosystem
carbon assimilation. The nitrogen component of JULES is described in (Wiltshire et al., 2021). In brief, JULES-ES represents

all the key terrestrial N processes. Inputs to the land surface are via biological fixation, fertilization and nitrogen deposition, with losses from the land surface occurring via leaching and gas loss, with Nitrogen deposition being externally provided to the model. JULES simulates a nitrogen-limited ecosystem by reducing the carbon use efficiency if there is insufficient available N to satisfy plant N demand. This results in a reduced Net Primary Production (NPP) when we include nitrogen limitation. The soil biogeochemistry is based on the representation of the four-pool RothC soil carbon (Clark et al., 2011) consisting of decomposable plant material (DPM), resistant plant material (RPM), microbial biomass (BIO), and humus (HUM). For each soil carbon pool there is an equivalent soil nitrogen pool (Wiltshire et al., 2021). Nitrogen transfers between the organic and inorganic nitrogen pools depend on decomposition rates and the C to N ratio of the organic pool.

Another important change is the inclusion of a fire module. Fire is simulated in JULES by the fire model INFERNO (INteractive Fire and Emission algoRithm for Natural envirOnments; (Mangeon et al., 2016)). Burned area is calculated from flammability and ignitions. Temperature, saturation vapour pressure, relative humidity, precipitation, together with soil moisture and fuel load from JULES give flammability by PFT, prescribed population density (Goldewijk et al., 2017) gives human ignitions, and prescribed climatological lightning gives natural ignitions. Previous evaluation of JULES-INFERNO, which compared model output against observational constraints, showed that burnt area is relatively insensitive to changes in lightning (Mangeon et al., 2016; Burton et al., 2022). Here we use INFERNO coupled to the dynamic vegetation model TRIFFID (Burton et al., 2019), enabling carbon cycle feedbacks from fire onto the land surface via vegetation mortality, regrowth, and burnt litter fluxes. Recent updates to INFERNO allow fire mortality to vary by PFT, and updates to the representation of land-use and PFTs in JULES allows for reduced burning in C3 Crop and C4 Crop PFTs (Burton et al., 2020), based on global trends of agricultural fire suppression (Bistinas et al., 2014; Andela et al., 2017). Background mortality rates where reduced compared to the model without fire to account for the extra vegetation mortality from fire, as per Burton et al. (2019). We use fire and background mortality in (Burton et al., 2020). C3 Pasture and C4 Pasture burning remain as per Burton et al. (2019), which reflects observational constraints that show an increase in burnt area in pasture areas (Kelley et al., 2019; Bistinas et al., 2014).

## 2.2 Modifications for ISIMIP2b

In the ISIMIP JULES-ES configuration the TRIFFID period, this is a number (in days) that defines the frequency that the TRIFFID dynamic vegetation code is called, has been reduced from a 10-day to a 1-day period in order to allow a shorter restart periods necessary to meet the diagnostic requirements of ISIMIP (a large number of variables on short temporal scales). A daily TRIFFID period also allows the vegetation dynamics to respond more realistically to variations in plant productivity on shorter timescales, although the effect of this change is minimal in test historical runs.

Another key difference to the standard setup of JULES is the use of daily meteorological driving data. JULES needs a model timestep of no more than 1 hour to accurately simulate the diurnal cycle and exchange of heat, water and momentum and avoid numerical instabilities. In the ISIMIP experimental setup, we use the internal disaggregator (Williams and Clark, 2014) to calculate driving data values at the model timestep of 1 hour. The method uses the IMOGEN model disaggregator (Huntingford et al., 2010) to initially disaggregate to 3-hourly, which the model linearly interpolates to 1 hour. The diurnal cycle of

downward shortwave radiation is calculated from the position of the sun in the sky. Temperature is calculated from a sinusoidal function with a maximum 0.15 of a daylength after local noon, normalised by the diurnal temperature range. Downward longwave radiation is a linear function of temperature, and specific humidity is kept below saturation at each timestep. Precipitation is considered to occur in a single event, with a globally specified 'duration' parameter (6h for convective rainfall, 1h for large-scale rainfall, convective snowfall and large-scale snowfall). Convective rainfall occurs when temperature exceeds

288.15°K. The model assumes convective rainfall is more intense and so leads to more runoff and less infiltration into the soil. Given that precipitation events do not, by construction, overlap with midnight GMT, on average, this produces a spurious trapezoidal diurnal cycle, which is zero at midnight GMT (Williams and Clark, 2014). Precipitation above 350 mm/day is redistributed. Note that convective precipitation occurs only on a fraction of the grid box (Best et al., 2011), set to 30% in the ISIMIP2b runs, and within this fraction is modelled as a negative exponential distribution (Johannes Dolman and Gregory,

1992). Therefore, the grid box average intensity is not the same as the effective intensity at a point. Given the strong effect of intensity on canopy interception and runoff, the water cycle in the model is sensitive to the duration parameter choices (Williams and Clark, 2014).

### 2.3 ISIMIP2b protocol

The ISIMIP2b experiments focus on understanding different levels of mitigation. They use two RCP scenarios to explore the international commitments made under the Paris Agreement to stabilise global warming at well below 2°C, relative to pre-industrial mean temperatures. ISIMIP2b uses simulations from the Coupled Model Inter-comparison Project 5 (CMIP5), using an historical scenario (1860-2005) and the RCP2.6 and RCP6.0 concentration pathways (post 2005) to represent a higher ambition, lower temperature outcome and a low ambition pathway respectively (Riahi et al., 2017). Land-use data and

population density are based on the Shared Socioeconomic Pathway (SSP2) scenario and applied to RCP2.6 and RCP6.0 simulations. Lightning ignitions for fire are from the LIS/OTD version 2.3.2015 climatology (Cecil, 2006), and applied for the historical and future as per Rabin et al. (2017) and Hantson et al. (2020). To capture a range of climate sensitivities, four CMIP5 GCM driving models are chosen: GFDL-ESM2M, HadGEM2-ES, IPSL-CM5A-LR and MIOC5. The GCM driving data is global bias-corrected daily data at 0.5° resolution (Hempel et al., 2013; Frieler et al., 2017; Lange, 2018). Bias-

correction means we can compare the observed and simulated impacts during the historical reference period with a smooth transition into the future period. The bias-correction methodology adjusts multi-year monthly mean distribution throughout the historical and RCP periods based on comparison of GCM output against EWEMBI reanalysis data distributions (Dee et

al., 2011), for the period 1979 to 2013, using CMIP5 RCP8.5 post-2005 (the end of CMIP5 "historic" period), such that trends and inter-annual variability are preserved in absolute and relative terms for temperature and non-negative variables respectively

(Lange, 2018). EWEMBI combines data from multiple reanalysis source to cover all required variables. The variables bias-corrected for ISIMIP2b are listed in Table 1 reproduced from Frieler et al. (2017). The ISIMIP2b bias correction includes humidity as well as shortwave and longwave radiation using quantile mapping. Transfer functions are used to adjust the distributions of daily anomalies from monthly mean values. ISIMIP2b bias-correction methods adjust distributions independently for each variable, grid cell and month, preserving the statistical dependencies between variables, in space and

changes over time. The bias correction approach preserves the trends (and therefore sensitivities) from different GCMs but removes absolute biases found over the reference period from the historical and RCP periods. Each GCM therefore has a different variability and simulated climate outside of the reference period, but with a smooth transition going from historical period into the reference period, or from the reference period into the future. Some small biases remain after bias-correction, particularly in precipitation (Figure S1), where biases are a small fraction of total local rainfall but can affect precipitation

particularly in the South America (Figure S2). As part of the setup provided here, we include code for preparing JULES data for submitting to ISIMIP and ensuring it conforms to the strict protocols (see Text S1) and the ILAMB system for rapid evaluation of the simulations (see Text S2).

**Table 1 Bias corrected variables for ISIMIP2b simulations, reproduced from Table 1 of Frieler et al. (2017)**


| Bias corrected variable | Unit | Source dataset over land | Source dataset over Ocean |
|---|---|---|---|
| Near-surface relative humidity | % | E2OBS | E2OBS |
| Near-surface specific humidity | kg kg$^{-1}$ | E2OBS | E2OBS |
| Precipitation | kg m$^{-2}$ s$^{-1}$ | WFDEI-GPCC | E2OBS |
| Snowfall flux | kg m$^{-1}$ s$^{-1}$ | WFDEI-GPC | E2OBS |
| Surface air pressure | Pa | E2OBS | E2OBS |
| Sea-level pressure | Pa | E2OBS | E2OBS |
| Surface downwelling longwave radiation | W m$^{-2}$ | E2OBS-SRB | E2OBS-SRB |

| Surface downwelling shortwave radiation | W m$^{-2}$ | E2OBS-SRB | E2OBS-SRB |
|---|---|---|---|
| Near-surface wind speed | m s$^{-1}$ | E2OBS | E2OBS |
| Near-surface air temperature | K | E2OBS | E2OBS |
| Daily maximum near-surface air temperature | K | E2OBS-ERA | E2OBS |
| Daily minimum near-surface air temperature | K | E2OBS-ERAI | E2OBS |

## 2.4 Model Evaluation

We evaluate the model for key impacts sectors, and use the International Land Model Benchmarking (ILAMB) tool (Collier et al., 2018) to assess model performance for GPP, ET, runoff and albedo. ILAMB evaluates performance against observations
from remote sensing, reanalysis data and fluxnet site measurements and produces graphical and statistical scores of model results. The model values and the observation datasets we use for the evaluation are given in Table 1. As ILAMB doesn't include vegetation cover evaluation, we also include the Manhattan Metric (Kelley et al., 2013) comparison with ESA CCI Land Cover tree, shrub, wood and grass cover (Harper et al., 2022). Results and further details of the ILAMB and vegetation cover analysis are provided in Text S2. We evaluate the historical simulations separately for each GCM because the bias
correction preserves inter-annual differences between GCMs. We conduct evaluation over common time periods between observations and simulation using the historical and, for observational periods beyond the end of 2006, RCP6.0.

**Table 2 the observations used to evaluate the JULES-ES fields**

| Model value | Observation for evaluation |
|---|---|
| GPP | The upscaled fluxnet product from (Jung et al., 2011) |
| ET | GLEAM (Miralles et al., 2011) and MODIS (Mu et al., 2011) estimates |
| Runoff | (Dai, 2021)(Dai, 2021) |
| Albedo | GEWEX SRB radiation observations (Stackhouse et al., 2011) |
| Burnt area | Global Fire Emission Database version 4 with small fires (GFED4s; (Van Der Werf et al., 2017) |

| Vegetation cover | ESA CCI Land Cover tree, shrub, wood and grass cover (Harper et al., 2022) |
|---|---|

## 2.5 The experimental suite

To run JULES we collect all the tasks and input files that are needed together into what is known as a suite; this allow runs to be reproduced using the exact same applications, options and commands as previously run possibly by another user, scheduling them to run based on the dependencies between the tasks. Full instructions on how to run JULES in a suite are provided on GitHub (https://jules-lsm.github.io/tutorial/bg_info/tutorial_julesrose/jr_structures.html#jrsuite). We provide a full setup for running the ISIMIP2b simulations using JULES-ES in the form of the suite u-cc669 available via the Met Office Science Repository Service (MOSRS - https://code.metoffice.gov.uk/trac/roses-u see data availability section for information). The bias corrected driving data is available from ISIMIP at https://www.isimip.org/gettingstarted/input-data-bias-adjustment/. We also use the following datasets from ISIMIP. Where preprocessing of these is required to use this data within JULES, this preprocessing code is also part of the suite.

- $CO_2$ concentration
- Future land-use patterns
- Nitrogen deposition
- Land-sea mask

The Total Runoff Integrating Pathways (TRIP) river routing model allows JULES to collect and route water through river channels, essentially converting runoff to river discharge or river flow. A TRIP 0.5° river routing ancillary file is also required for these runs, which is available from http://hydro.iis.u-tokyo.ac.jp/~taikan/TRIPDATA/.

## 3 Model evaluation

### 3.1 Water

We evaluated the water cycle using runoff derived from (Dai, 2021) for the 50 largest river catchments. Estimated observed mean basin runoff combines river flow measurements at downstream gauge stations with a river flow model to estimate the flow at the river mouth. By assuming there are no losses from the river, we calculated the long-term mean, basin averaged runoff by dividing the river flow at the river mouth by the basin area.

Particularly in temperate regions north and south of the equator, simulations using all 4 sets of driving data show similar biases. However, differences between the driving datasets are greatest in tropical or sub-tropical river catchments (Figure 1). This is particularly evident in the Amazon basin, where mean runoff biases range from approximately 0 in the simulation driven by

HADGEM2-ES, to more than -0.6mm day$^{-1}$ in the simulation driven by IPSL-CM5A-LR. Strong variations between the simulations are also seen in the Brahmaputra basin with some smaller variations in the Chang Jiang basin (Figure 2). The

general underestimate of runoff in the higher latitudes may be due to the treatment of moisture infiltration into partially frozen soils (see below) but could also be caused by biases in precipitation estimates due to the gauge "under-catch" of snow (Adam and Lettenmaier, 2003). In arid and semi-arid basins river flow and runoff tends to be over-estimated, which could be due to missing processes such as river channel evaporation and transmission losses (Haddeland et al., 2011; Döll and Siebert, 2002) and anthropogenic water extraction, primarily irrigation (Richey et al., 2015) .


Figure 2 compares the long-term monthly mean river flow (1980-2014 inclusive where observations are available) over the six largest rivers to those of downstream observations in (Dai, 2021). All the simulations reproduce the overall seasonal river flow over the Amazon. After the wet season, the modelled river flows decline earlier than observed and the simulated river flows at "low water" are too low. This could be due to too much evaporation in the drier months (see JJA in Figure S3) and/or

the simulated speed of flow through the soil and/or river channel being too fast. Though GFDL-ESM2M and IPSL-CM5A-LR driving precipitation data also has a dry bias during the dry season (Figure S2). All simulations over-estimate river flow over the Congo, mainly due to over-estimates in the rainy months. This could be driven by too little evaporation from the vegetation canopy or from flooded areas (See ET in Figure 3 and Figure S3). The simulations reproduce the seasonal river flow over the Orinoco well. The timings of peak river flow for the Chiangjiang and Brahmaputra are well simulated although the amplitudes

are too low. Dams, which we do not model, are likely to affect the observed seasonal cycle. The simulated river flow over the Mississippi lags observations by several months. This lag is also evident over many high latitude basins (not shown). The observed river flow peak is mainly driven by Spring melt, whereas the simulated river flow peak is in line with that of precipitation. This is due to this configuration allowing significant soil infiltration of snow melt when the soil surface is mainly frozen, rather than resulting in surface runoff. This also may result in the underestimate of annual river flow because once the

water has infiltrated the soil it may then be evaporated.

### 3.2 Surface fluxes

Global Gross Primary Productivity (GPP) is 134-137 PgC/yr, depending on driving data, which is above the estimate of IPCC AR6 (Canadell et al., 2021) of 113 PgC yr-1, but agrees well with the estimates of 146 ± 21 PgC/yr of (Cheng et al., 2017);

Table S1). Net Biome Productivity (NBP) is 0.94-1.46 PgC/yr between 2011-2020, within the 1.0-2.8 PgC/yr range estimated by the Global Carbon Budget (Friedlingstein et al., 2022). All simulations show positive GPP biases in similar regions, such as central and southern Africa, south of the Himalaya and east towards Bangladesh and Myanmar compared to (Jung et al., 2011) observations (Figure 3). South America is a more complicated picture with Brazil broadly split between negative GPP to the northeast and positive GPP to the country's southeast. For Brazil, the ET bias has a more northwest – southeast split,

with the northwest having a slightly negative bias and the southeast more positive. The northwest bias in ET and the bias in

GPP in South America is more prominent and wide-spread in early wet season (September-November) when driven by climate data from GFDL-ESM2M and IPSL-CM5A-LR (see Figure S4) and is due to a longer dry season in both sets of driving data (Figure S2), with rains starting in October rather than September. The far north of Columbia, Bolivia and Argentina also have a negative bias in both GPP and ET across the simulations. Australia also has a north-south split with a slight positive ET bias to the north and the inverse to the south.

Albedo (Figure 3, right column) generally shows a positive bias across most regions and simulations. However, there are small regions with a negative bias, for example, south of the Sahel and small regions at higher northern latitudes. Eastern Siberia has a positive bias in all simulations (see Discussion).

### 3.3 Biomes

All simulations show similar vegetation cover patterns that largely follow observations, capturing high tree cover fractions in boreal and tropical forests, grass cover in tropical, temperate and boreal grasslands and bare ground in arid regions (Figure 4). There are, however, some biases common to all simulations. Tree cover fraction is too high globally, with a simulated range of 4.97-5.31Mkm$^2$,which is higher than observations and depends on driving data (Table S2). Shrub and grasses dominate eastern Siberian Taiga in the model instead of observed high tree cover (Figure 4). There is also slightly too much shrub cover in tropical forests at the expense of tree cover, contributing to a global bias in shrub cover of 1.22-1.31 Mkm$^2$. Conversely, simulated tropical tree cover is too high in savanna regions, giving the impression of more continuous and less fragmented forests across the tropics (Figure 4). Boundaries between temperate and warm temperate woodlands/forests and tropical forests are too sharp, suggesting JULES-ES does not capture processes in temperate woodland transition. Savanna and grasslands tend to be too narrow, with more bare soil in the models in semi-arid regions such as southern Africa, the Mojave desert and the Sahel.

### 3.4 Fire

In addition to the simulations without fire submitted to the archive, we performed additional simulations with fire and fire feedbacks switched on. These fire simulations provide burnt area and alter vegetation cover, carbon, fluxes, albedo and runoff (Burton et al, 2019). Burnt area is similar across all four simulations  (Figure 5) . The model simulates present-day burnt area well compared to satellite observations, with the global total burnt area average for 2000-2020 observed by MODIS CCI v5.1 as 4.55 Mkm$^2$, and the model simulating between 3.94-4.43 Mkm$^2$ depending on driving data (Table S3). The model captures the high burnt area in southern hemisphere Africa - a common area of low bias in global fire models (Hantson et al., 2020). The model also performs better than other FireMIP models at simulating the high burnt areas in northern hemisphere Africa , though fire is still lower than in observations. This is partly due to very low simulated burnt areas in Nigeria's Guinean savanna,

which (Kelley et al., 2019) show is a consequence of global parameterisations of population density and agricultural drivers of burnt area.


We also show too low burnt area in Australia, which is a common problem across fire models (Hantson et al., 2020), even in models optimized to burnt area observations (Kelley et al., 2019; Bistinas et al., 2014). This may be due to the unique fire ecology in Northern Australia (Kelley, 2014) and to high uncertainty in observations of burnt area (Giglio et al., 2010). High burning in South America occurs in areas where cropland fragmentation reduces burnt area beyond the extent of agricultural

areas (Kelley et al., 2019; Andela et al., 2017), which is hard to reproduce in tile and PFT-based models (Hantson et al., 2016). We also simulate too-low burnt area in Eurasia. Some of this observed burnt area at these high latitudes comes from peatland fires, which, as in most other fire models (Rabin et al., 2017), are not simulated in INFERNO. Observational burnt area products tend to underestimate levels of burning in forests (Randerson et al., 2012), which may explain the slight bias towards too-high simulated burnt area in tropical forests.


Fire is simulated well in savanna bands (15 degrees north and south), which improves the representation of tree cover by reducing the positive bias compared to observations (Figure 6; Table S2).  The inclusion of interactive fire has the overall effect of decreasing simulated global tree cover (from 38.66-39, depending on driving data to 34.32-35.65 Mkm$^2$) to more in line with observations (34.86 Mkm$^2$) and increasing grasses and bare soil. In the tropics, this tends to bring the modelled

"forest" (areas dominated by trees) more in line with the observations, with high simulated tree cover restricted to observed forested areas. However, including fire does reduce tree cover in Savanna areas (Figure S5), particularly in Africa, which means the spatial distribution of trees in this region compared with observations is not as good as without fire  (Table S2). There is an increase and improvement in tree cover in some high latitude North America due to changes in background mortality in the with fire simulation. Fire reduces shrub cover to well below observations (Table S2), though given the well

documented issues in distinguishing tall and short woody vegetation (Gerard et al., 2017; Adzhar et al., 2022), it is probably more meaningful to assess total woody cover. Here, including fire reduces model bias (from 6.19-6.59 to -3.60- -2.25 Mkm$^2$) and improves the spatial pattern (Table S2). Including fire reduces the global bias of high GPP when compared against estimated by the Global Carbon Budget (Friedlingstein et al., 2020)., bringing global total down by ~2 PgC/yr across all simulations (Table S1). However, fire does slightly degrade GPP's spatial pattern (Table S1). Including fire also reduced global

NBP (Table S1) by 0.12-0.38 PgC/yr, depending on driving data.

Fire alleviates some of the high bias in ET, improving the model's overall performance (Table S4). Without fire, in semi-arid areas we already over-estimate river flow (in part probably due to human extraction). The addition of fire lowers ET, thereby increasing river flow bias in semi-arid regions, which slightly degrades overall runoff performance (Figure S6). Fire also

improves spatial pattern of albedo (Table S5), though seasonal performance decreases. This is in line with well-documented biases in fire seasonal cycles across all global fire models, which tend to have longer than observed fire seasons in tropical

savannas, and, in human dominated fire regimes, season timing shifted and often late compared to observations (Hantson et al., 2016, 2020). Previous evaluations of JULES configurations incorporating INFERNO also show these biases (Burton et al., 2019, 2022, submitted; Hantson et al., 2020).


## 4 Discussion

We have presented simulations of the JULES-ES land surface model, run according to the ISIMIP2b protocols using bias-corrected climate model data from 4 GCMs for the historical period. The configuration will be used to perform simulations under 2 future scenarios (RCP2.6 and RCP6.0). The JULES-ES ISIMIP2b configuration simulates the surface fluxes (GPP,

ET, albedo) reasonably well. Including fire improves the ET and albedo, but not the GPP which is biased high. Including fire in the simulations currently degrades runoff (Table S6).

The configurations of JULES can capture the seasonal cycle of many of the largest rivers, although high latitude rivers and managed rivers are generally not captured as well. Including irrigation and structural hydrological developments, such as dams

and reservoirs, would likely improve the simulations of managed rivers. Previously, (Falloon et al., 2011) found that GCM precipitation biases contribute to errors in TRIP river flows for some basins in both HadGEM1 and HadCM3. In this study, we use bias-corrected data which reduces these errors, meaning that differences in JULES-ES results between the driving models are due to differences in inter-seasonal or inter-annual variability between driving models (e.g. Figure S2). However, errors in evapotranspiration, runoff generation or other missing processes e.g., snow accumulation and snow melt processes,

could also contribute. Uncertainty in precipitation due to the sparsity of observation networks and the under-catch of solid precipitation for high latitude (Falloon et al., 2011) and altitude rivers e.g., in the Himalaya (Mathison et al., 2015), means that it is difficult to interpret model performance in these basins.

Some basins show the same bias direction in runoff (Figure 1) and ET (Figure 3), notably that the Amazon is too dry for both

variables and the Nile too wet, whereas we would expect opposing biases if they were from land surface simulation. In these basins, there is a dry and wet bias (respectively) in the driving precipitation data (Figure S1). HadGEM2-ES and IPSL-CM5A-LR have particularly dry driving data in the Amazon, and this results in the driest runoff and ET in the simulation. This translates to biases in GPP (Figure 3) and vegetation cover (Figure 4). So, while ISIMIP bias-correction reduces climate model biases compared to those in an Earth System Model (see evaluation in (Sellar et al., 2019)) or when run with a non-bias

corrected climate (Burton et al., 2022), they are not eliminated.

Land cover is an important factor for the surface fluxes. Grassy regions for example, correlate with the regions of positive ET bias. The annual mean global GPP biases are small, but this is not the case for GPP on a seasonal timescale and over a smaller

region such as South America. The albedo and land cover area bias are also closely related. For example, if JULES simulates a larger number of trees than observed, this may lead to a lower albedo than observed, and vice versa. Conversely, higher grass cover just north of the Sahel corresponds to simulated low-albedo biases. Vegetation impacts on albedo are particularly important at high latitudes where there is snow cover; for example the positive albedo bias in Eastern Siberia is because JULES simulates too few trees and too much grass there. Grasses are more readily buried below snow than trees, making these areas more reflective (Sellar et al., 2019), which in turn affects the albedo. JULES represents the bending and partial burying of vegetation by snow (Ménard et al., 2014); however the settings controlling this interaction described in (Sellar et al., 2019) have been tuned for the coupled UKESM1 model rather than the standalone JULES model. Eastern Siberia is a vulnerable region which has experienced increased climate-related impacts, including heatwaves (Ciavarella et al., 2021) and fires (Kelley et al., 2019); it is very likely that climate change will exacerbate these feedbacks by the end of the 21st century (United Nations Environment Programme, 2022). Developments by Mercado et al. (2018) which improve the representation of plant acclimation to thermal stress may improve spatial variations across different vegetation types in JULES.

In general, the simulations with fire improve the vegetation and productivity distribution, particularly the distribution of tropical forests, the boundary between forests and savannas, and in North America. Developing JULES to include fire processes will improve simulations for these areas and properly capture the climate impacts on vegetation cover and carbon fluxes. The results show that there are too few trees compared to observations for western parts of Brazil. The simulations with fire on improve tree cover in savanna, which is consistent with the findings of (Staver et al., 2011; Lasslop et al., 2016), however there is still ongoing discussion around how much impact fire really has on tree cover in the savanna compared to other dry disturbances such as wind throw, heat stress, rainfall distribution (Veenendaal et al., 2018; Brovkin et al., 2009).

## 5 Conclusions

We have presented a configuration of JULES-ES set up to run and generate output following the ISIMIP protocols. We provide a suite for running the simulations that includes driving data, ancillaries, postprocessing and first look evaluation (ILAMB) for any phase of ISIMIP. Outputs using this set up were submitted to the biomes and water ISIMIP2b sectors, and our evaluation helps inform any difference between JULES-ES and other models participating in ISIMIP2b. The suite also provides a starting point for further JULES-ES developments. We evaluate a set up with representation of fire using the INFERNO fire model in anticipation of ISIMIP3 which will include a Fire Sector. We show that including fire has an impact on model results and is important to include in simulations of climate impacts. While fire mostly improves model performance, it does degrade certain vegetation distributions (for example by simulating too little larch forest) and runoff. However, fire has a substantial impact on both ecosystem composition and hydrological processes and should therefore still be included when studying impacts under changing climate and environmental conditions. Therefore, while documentation of the configuration

without fire will be useful for anyone using previously submitted results, we recommend using the configuration with fire in future JULES-ES development. Future work using this configuration and new phases of ISIMIP will focus on using the full benefit and extent of the ISIMIP ensemble to enable more in-depth exploration of climate impacts together with the quantification of earth system uncertainties, co-benefits of mitigation and adaptation to climate change.


**Data and code availability**

The JULES-ES for ISIMIP configuration (based on JULES version 5.5) is preserved at https://code.metoffice.gov.uk/trac/roses-u/browser/b/k/8/8/6 (fire off) and https://code.metoffice.gov.uk/trac/roses-u/browser/c/f/1/3/7 (fire on). JULES and associated configurations are freely available for non-commercial research use as set

out in the JULES user terms and conditions (http://jules-lsm.github.io/access_req/JULES_Licence.pdf). For a comprehensive guide on how to access, install and run the configurations used in this research, we direct the reader to Appendix A of (Wiltshire et al., 2020) available at https://gmd.copernicus.org/articles/13/483/2020/#section6. Note that to view and use the JULES-ES source code, access will be required to the Met Office Science Repository Service (https://code.metoffice.gov.uk/trac/home) and is available to those who have signed the JULES user agreement. The easiest way to access the repository is by completing

the online form to register here: http://jules-lsm.github.io/access_req/JULES_access.html.

The data and code used for the evaluation of the JULES-ES outputs with iLAMB in the study are available at https://www.ilamb.org/datasets.html and https://github.com/rubisco-sfa/ILAMB with a BSD 3-clause "New" license (https://github.com/rubisco-sfa/ILAMB/blob/master/LICENSE.rst)

The JULES model data output used in the model evaluation in the study are available at https://data.isimip.org/, using the search tag 'jules-es-55' https://data.isimip.org/search/query/jules-es-55/ with a Creative Commons Attribution 4.0 International license (https://creativecommons.org/licenses/by/4.0/)

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

**Figures**

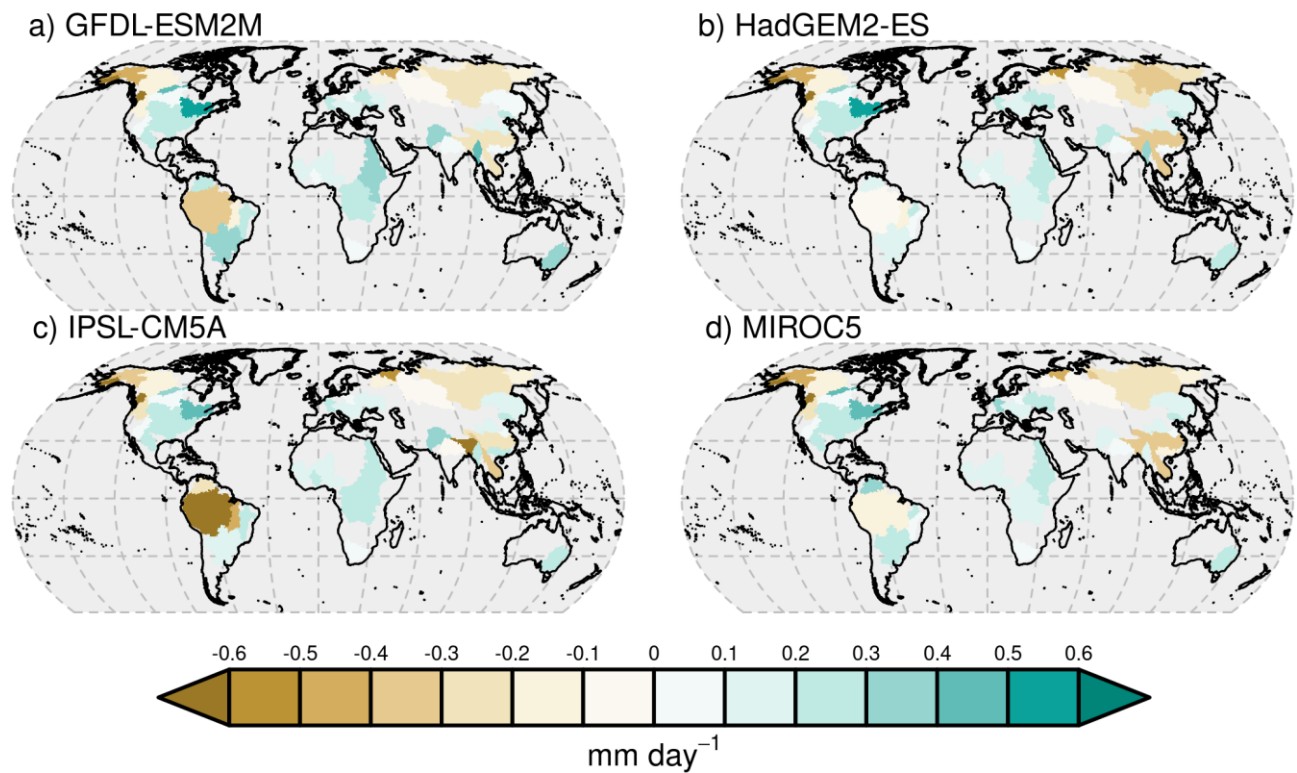


**Figure 1: Multi-year mean bias of catchment scale runoff simulated by JULES driven by 4 sets of climate driving data compared to runoff derived from** (Dai, 2021)**. Number of years of observations contributing to the multi-year mean varies depending on catchment, and the observations that are available. Observations used are within the period 1980-2006 . ISIMIP2b forcing data**
**derived from 4 CMIP5 GCMs: GFDL-ESM2M; HadGEM2-ES; IPSL-CM5A-LR; MIROC5.**

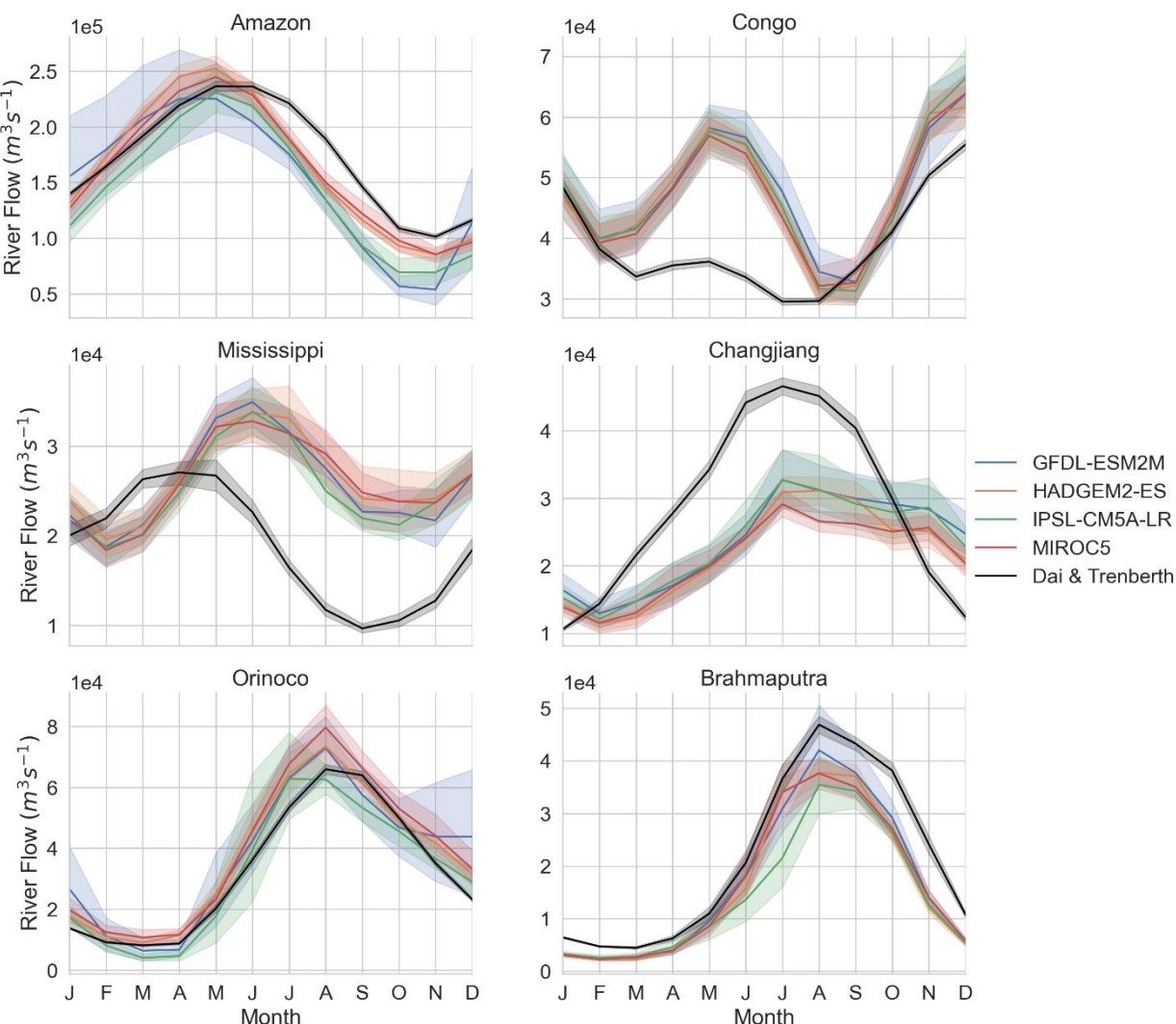

**Figure 2: Comparison of the simulated long term monthly mean river flow with observations** (Dai, 2021) **for the six largest rivers: a) Amazon (Obidos, -1.95N,-55.51E); b) Congo (Kinshasa, -4.3N, 15.3E); c) Orinoco (Pte Angostu , 8.15N,-63.6E); d) Changjiang (Datong, 30.77N, 117.62E); e) Brahmaputra (Bahadurabad, 25.18N, 89.67E); f) Mississippi (Vicksburg 32.31N, -90.91E). The observations are over the time period 1980 to 2010.**


s

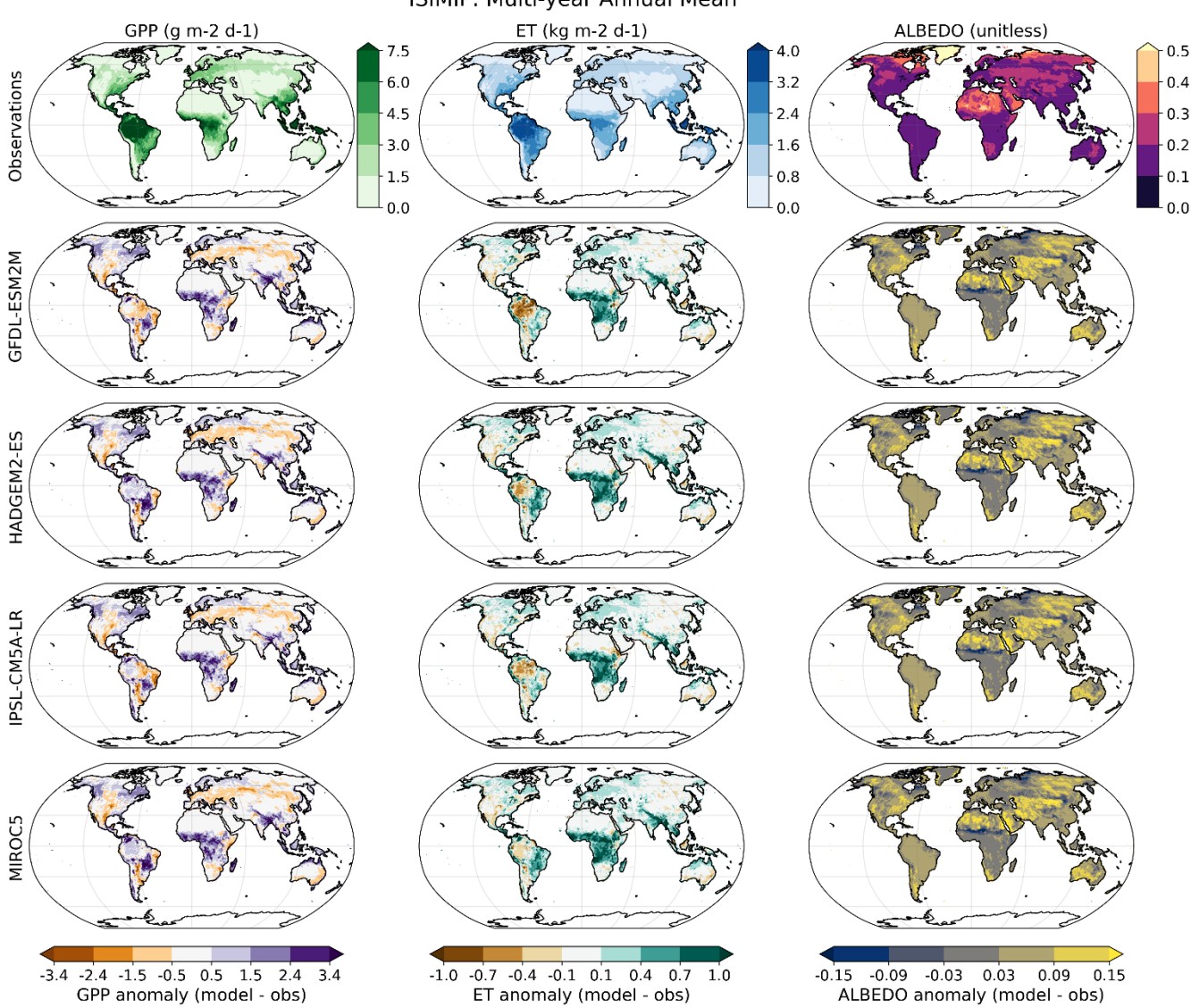


**Figure 3: Multi-year annual mean for GPP (column 1), Evapotranspiration (Column 2) and Albedo (column 3) for observations (row 1) and subsequent rows show the anomaly compared with observations for each set of ISIMIP2b forcing data derived from: GFDL-ESM2M (row 2), HADGEM2-ES (row 3), IPSL-CM5A-LR (row 4) and MIROC5 (row 5). Observations are downloaded from iLAMB (https://www.ilamb.org/doc/ilamb_fetch.html) and the datasets shown are GBAF for GPP** (Jung et al., 2011)**, GLEAM**
**for ET (**(Miralles et al., 2011) **and GEWEX.SRB for Albedo** (Stackhouse et al., 2011)**.**

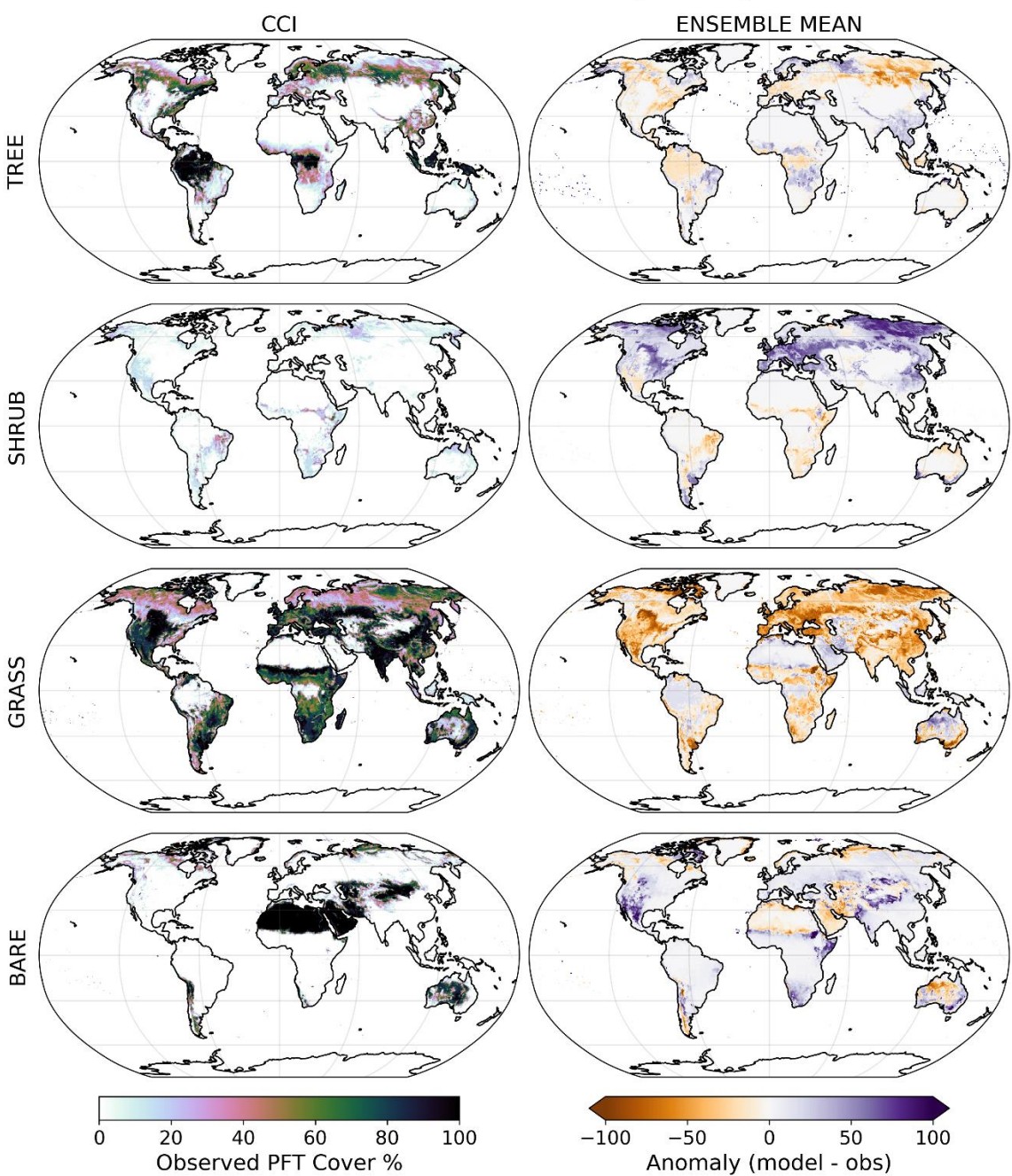

**Figure 4: Observed vegetation fractional cover, derived from ESACCI Land Cover (v2.0.7) for 2010 (K. L. Harper et al., 2022), for (left to right) tree, shrub, grass and unvegetated (bare) fraction. Subsequent rows show the difference between model (without fire) and ES CCI observations for each set of driving data.**


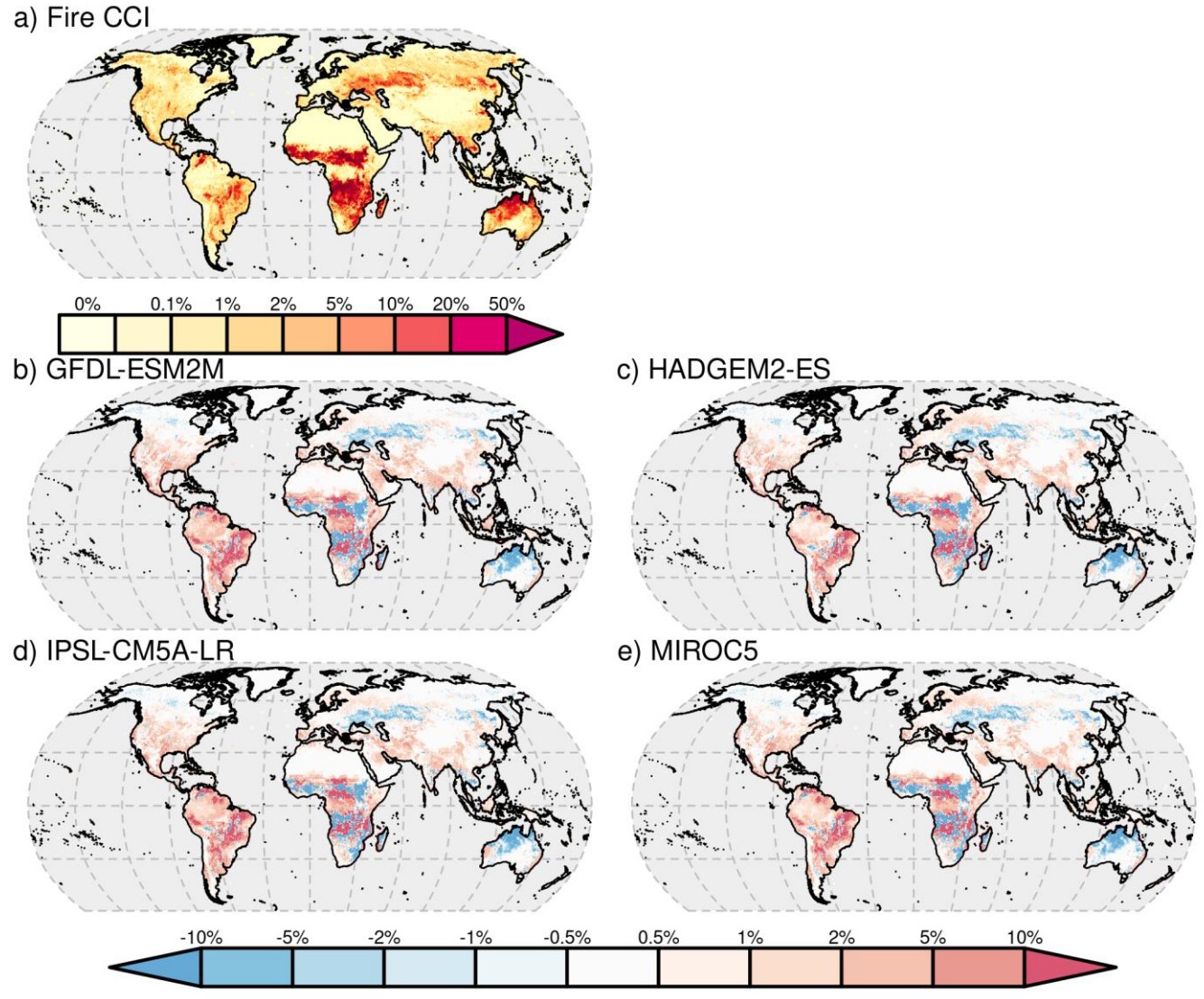

**Figure 5: Present day percentage burned area (2000-2020) from and Fire CCI observations (a) and modelled by JULES driven by GFDL-ESM2M (b), HadGEM2-ES (c), IPSL-CM5A-LR (d), MIROC5(e)**

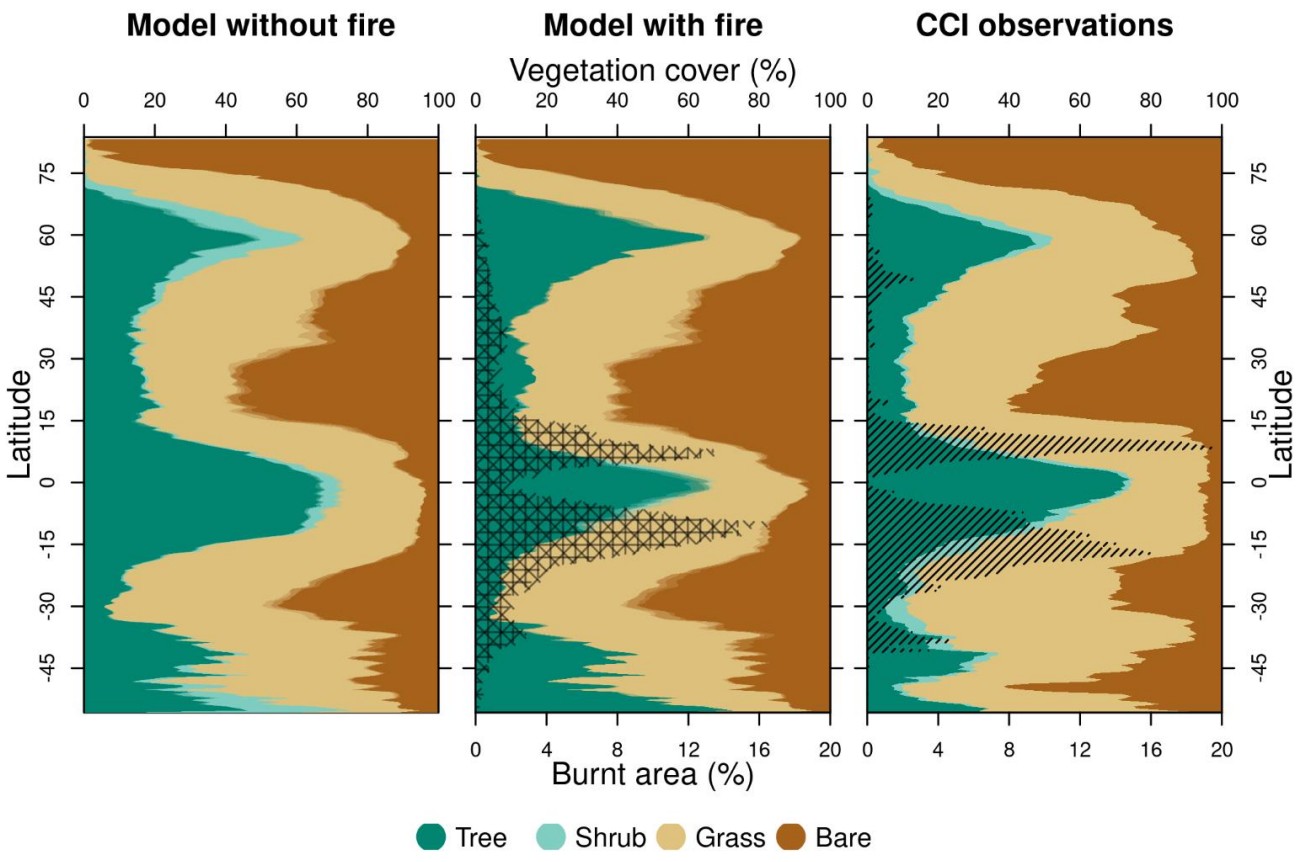

**Figure 6: Modelled vegetation cover without fire (left) and with fire (middle) compared to observations from ESA CCI Land Cover and Fire (right). Dark green, light green, light brown, dark brown indicates tree, shrub, grass and unvegetated fraction of the latitude band. Shaded transition between colours indicates ensemble range, which is quite narrow indicating agreement across ensemble members. Black hashing indicates burnt area, with observations taken from MODIS CCI v5.1** (Chuvieco et al., 2018)**. In "Model with fire", burnt area from the 4 driving models is shown by hatching at 4 different angles.**

## Author contribution

Camilla Mathison: Conceptualisation; Data curation; Formal analysis; Investigation; Methodology; Project administration; Software; Writing – original draft; Writing – review & editing

Eleanor Burke: Conceptualisation; Data curation; Investigation; Methodology; Software; Writing – original draft; Writing – review & editing

Andrew Hartley: Data curation; Formal analysis; Investigation; Methodology; Project administration; Validation; Visualisation; Writing – original draft; Writing – review & editing

Douglas I Kelley: Formal analysis; Investigation; Methodology; Validation; Visualisation; Writing – original draft; Writing – review & editing

Chantelle Burton: Data curation; Methodology; Formal analysis; Investigation; Validation; Visualisation; Writing – original draft; Writing – review & editing

Eddy Robertson: Formal analysis; Software; Validation; Writing – review & editing

Nicola Gedney: Validation; Visualisation; Writing – original draft; Writing – review & editing

Karina Williams: Methodology; Writing – review & editing

Andy Wiltshire: Writing – review & editing

Richard J Ellis: Software; Writing – review & editing

Alistair A Sellar: Software

Chris D. Jones: Validation; Writing – review & editing

## Acknowledgments

Work and its contributors were supported by the Newton Fund through the Met Office Climate Science for Service Partnership Brazil (CSSP Brazil) (CB, CM, AH, AW CJ and NG), the Met Office Climate Science for Service Partnership South Africa (CSSP South Africa) (AH and CM). CM, KW, AW, AH, AS, EB, ER and CJ  were supported by the Met Office Hadley Centre Climate Programme funded by BEIS and Defra. The contribution by DIK and RJE was supported by the UK Natural Environment Research Council through The UK Earth System Modelling Project (UKESM, grant no. NE/N017951/1). DIK was additionally supported by UK Natural Environment Research Council as part of the NC-International programme (NE/X006247/1). EB and CJ were supported by the European Union's Horizon 2020 research and innovation programme under Grant Agreement No 101003536 (ESM2025 - Earth System Models for the Future)