# Peer review of "Description and Evaluation of the JULES-ES setup for ISIMIP2b"

_EGUsphere, 2022_

## Author Comment (AC1)

**Response to Reviewers**

Many thanks to both reviewers for their constructive comments on our manuscript. We have tried to address each below. Comments from the reviewers are in blue italics and our authors responses are in black.

**Reviewer 1**
**General comment:**
*The authors provide a description of the JULES-ES model used as part of the ISIMIP project. The paper is overall well -written and easy to follow, with a fairly detailed description of key model processes and the model set-up. The authors also present an evaluation of present-day model performance to indicate its skill in simulating key ecosystem and water cycle processes. It was particularly great to see the authors using the ILAMB evaluation tool for this purpose as it forms a useful basis for benchmarking any future JULES developments against JULES-ES in a transparent way. Overall, I do not have any major concerns about the manuscript and recommend it for publication subject to some minor revisions.*

**Authors response to general comments**
Thank you for your constructive comments. We have described how we will address your comments in the text below.

**Responses to specific comments:**

*L12: Best to spell out what ISIMIP2b stands for (a few other abbreviations in the abstract -GCMs and GPP- are not used and thus not needed)*

Noted, this will be corrected in the manuscript.

*L30: what do you mean by "representing impacts"?*

The JULES-GL7 is a configuration intended to be used as the land surface component of coupled land-atmosphere models in short-range NWP through to climate configurations, where the principal aim is to simulate biophysical land surface fluxes of energy, moisture and momentum back to the atmosphere. In contrast, the JULES-ES configuration also includes biogeochemical fluxes governing carbon and nitrogen cycles that influence earth system processes that might be considered more relevant to ecosystems and people, which could be affected by climate change. We will make this clearer in the text by modifying the sentence on line 30 as follow:

> *JULES-ES builds on the JULES-GL7 configuration described in* (Wiltshire et al., 2020) *by including additional biogeochemical fluxes governing carbon and nitrogen cycles that influence earth system processes; these are considered more relevant to ecosystems and people, which could be affected by climate change.*

*L39: Intrigued to know why JULES-ES is more efficient than the UKESM version? What are the main differences?*

This configuration of JULES is the same version used in the UKESM1 simulations. However, we run this in offline mode, which means the atmosphere is prescribed, making the JULES standalone faster to run than the fully coupled model. We will make this clearer in the text (see next comment).

L40: I think you mean uncertainty in climate forcing and its impact on land? Otherwise, I don't quite follow the "land surface forcing" terminology as you only use one LSM

Agreed will change to climate forcing in the text, so that the sentence reads as follow:

> An advantage of using JULES-ES as an offline impacts model is that it uses a prescribed atmosphere, without these feedbacks JULES-ES is more computationally efficient than the closely aligned land surface scheme coupled to the atmosphere in UKESM1 (Sellar et al., 2019a), and using multi-model climate ensembles sample scientific uncertainty in our understanding of the climate system that would not be possible within a single climate model framework.

L61: should this say "lower" carbon assimilation rates?

Yes, this will be corrected in the text

L65: C4 grasses twice

One of these should be C3 grasses, thanks for pointing it out. This will be corrected.

L82: would be good to briefly discuss how realistic it is to assume identical physiology?

Here, we use a scheme linked to TRIFFID described in Sellar et al. (2019) and Robertson (2019) because it allows the crops to be linked to the carbon and nitrogen cycles and represents agriculture's impact on vegetation distribution. Largely we keep the crops the same as grasses for consistency with the JULES-ES set up within UKESM1. However, developing new versions of the grasses that are more productive and like crops could be an option for future development of this crop model in JULES.

L109: Do you mean the TRIFFID time stepping?

The TRIFFID Period refers to the frequency of the call to the TRIFFID model, this is given in days. In this set up we have set this to 1. We will make this clearer in the text by adaptingthe sentence to read:

> In the ISIMIP JULES-ES configuration the TRIFFID period**, this is a number (in days) that defines the frequency that the TRIFFID dynamic vegetation code is called,** has been reduced from a 10-day to a 1-day period in order to allow a shorter restart periods necessary to meet the diagnostic requirements of ISIMIP (a large number of variables on short temporal scales).

L133: Suggest rewording "they are consistent" because it implies that the different ISIMIP2b scenarios are consistent with a >2deg world but this is not true for RCP6.0. I thus found the sentence a little confusing

Agreed, will modify this to say that they "explore" the international commitment made under the Paris agreement to stabilise global warming at well below 2°C.

L136: historical ends in 2005?

Thanks, this will be corrected in the manuscript

L142: should be post-2005?

Thanks this will be corrected in the manuscript.

L141-2: this reads like only the period 1979-2013 was bias-corrected?

We have corrected this sentence to clarify that bias-correction is applied throughout this historical and RCP periods, as follows:

> The bias-correction methodology adjusts multi-year monthly **mean distribution throughout the historical and RCP periods based on comparison of GCM output against EWEMBI reanalysis data distributions** (Dee et al., 2011) for the period 1979 to 2013, using CMIP5 RCP8.5 post-200**5** (the end of CMIP5 "historic" period), such that trends and inter-annual variability are preserved in absolute and relative terms for temperature and non-negative variables respectively (Lange, 2018).

L143: Reference should be in brackets

Thanks this will be corrected

L145: Need to mention all variables that are bias-corrected, this suggests only humidity etc. were

Agreed that this will make the text cleare. We will add the following to the text:

> EWEMBI combines data from multiple reanalysis source to cover all required variables. The variables bias-corrected for ISIMIP2b are listed in Table 1 reproduced from Frieler et al. (2017). The ISIMIP2b bias correction includes humidity as well as shortwave and longwave radiation using quantile mapping.

We also will add information for all variables that are bias-corrected for ISIMIP2b using Table 1 from Frieler et al. (2017).

**Table 1 Bias corrected variables for ISIMIP2b simulations, reproduced from Table 1 of Frieler et al. (2017)**

| Bias corrected variable | Unit | Source dataset over land | Source dataset over Ocean |
|---|---|---|---|
| Near-surface relative humidity | % | E2OBS | E2OBS |
| Near-surface specific humidity | $kg\ kg^{-1}$ | E2OBS | E2OBS |
| Precipitation | $kg\ m^{-2}\ s^{-1}$ | WFDEI-GPCC | E2OBS |
| Snowfall flux | $kg\ m^{-2}\ s^{-1}$ | WFDEI-GPC | E2OBS |
| Surface air pressure | Pa | E2OBS | E2OBS |
| Surface downwelling longwave radiation | $W\ m^{-2}$ | E2OBS-SRB | E2OBS-SRB |
| Surface downwelling shortwave radiation | $W\ m^{-2}$ | E2OBS-SRB | E2OBS-SRB |

| Near-surface wind speed | m s$^{-1}$ | E2OBS | E2OBS |
|---|---|---|---|
| Near-surface air temperature | K | E2OBS | E2OBS |
| Daily maximum near-surface air temperature | K | E2OBS-ERA | E2OBS |
| Daily minimum near-surface air temperature | K | E2OBS-ERAI | E2OBS |

L148-9: Again this reads like only the reference period was corrected, is this correct? Also a bit confused by each GCM having a "different variability and climate" outside of reference period, would this not also be true during the reference period? Perhaps consider rewording some of this section as I find it challenging to follow

We will modify this sentence as follows to clarify:

The bias correction approach preserves the trends (and therefore sensitivities) from different GCMs but removes absolute biases **found over** the reference period **from the historical and RCP periods.**

L163: check grammar

We wil correct the grammar.

L164: historic -> historical

Will modify in the manuscript

L170: could you very briefly explain what a suite is for those not familiar with JULES

We will add the following text to this section to explain what a suite is.

> To run JULES we collect all the tasks and input files that are needed together into what is known as a suite, this allow runs to be reproduced using the exact same applications, options and commands as previously run possibly by another user, scheduling them to run based on the dependencies between the tasks. Full instructions on how to run JULES in a Rose suite are provided on github (https://jules-lsm.github.io/tutorial/bg_info/tutorial_julesrose/jr_structures.html#jrsuite).

L179: Please explain what TRIP is

We will explain TRIP in the text as follows:

> **Total Runoff Integrating Pathways (TRIP) river routing model allows JULES to collect and route water through river channels essentially converting runoff to river discharge or river flow. A** TRIP 0.5° river routing ancillary is also required **for these runs, which is** available from http://hydro.iis.u-tokyo.ac.jp/~taikan/TRIPDATA/.

Figure 1: suggest reverting colours, red normally implies too little water

We will replace figure 1 with this plot with a more appropriate colour scheme:

[Figure]

There are smaller but still notable variations for the Chang jiang, which willl have note in the text.

L193: I don't quite follow how water extractions "disproportionally affecting groundwater depletion" leads to high runoff biases?

We cited Richey et al. (2015)  which primarily refers to ground water depletion. However, Döll and Siebert (2002) and Haddeland et al. (2011) shows water extraction, primarily through irrigation, also has a substantial effect on river runoff. We will therefore add these references and removed "disproportionally affecting groundwater depletion" to avoid confusion.

Also what about other model processes that might be missing, such as river channel evaporation and transmission losses?

We will add that these are important missing processes for arid/semi-arid basins and included an appropriate reference.

The sentence will read:

> In arid and semi-arid basins river flow and runoff tends to be over-estimated, which could be due to missing processes such as anthropogenic water extraction, primarily irrigation (Richey et al., 2015) and river channel evaporation and transmission losses (Haddeland et al., 2011; Döll and Siebert, 2002).

We will add units to the figure and caption.

Figure S3: monthly precipitation? And are the units mm/month?

We will update the units and added them to the caption

Figure S4: the panels and legend are far too small to read, and the righthand panels are cut off.

We will split the Figure S4 into separate plots for ET and GPP and update references to Supplemntary figures in the text. Seasonal albedo is not discussed so we shall remove this.

[Figure]

**Figure S4:** Seasonal variation in Evapotranspiration (ET) for tropical South America.

[Figure]

**Figure S5:** Seasonal variation in Gross Primary Production (GPP) for tropical South America

L205: is the ET bias shown anywhere? If so, refer to a figure

Figure 3 middle column shows the Evapotranspiration anomaly compared with observations shown in the first row, and Figure S4 shows seasnal ET biases over the Amazon. We shall make sure these are referenced in the appropriate passages on text.

Thanks, Corrected

Thanks, Corrected

L230-232: Any thoughts on what might be causing these albedo biases?

We discuss the causes for albedo biases in the part of the discussion that refers to Landcover, as indicated in the text. This is because it related to biases in other evaluated components, and we use the discussion to link different aspects of land surface evaluation together. However, we realise we omitted to mention causes of albedo biases in the Sahel, which we now add to the discussion:

> The albedo and land cover area bias are also closely related. For example, **if JULES simulates a larger** number of trees than observed, this may lead to a lower albedo **than observed or if JULES simulates a lower number of trees this may lead to a higher albedo than observed. Conversely, higher grass cover just North of the observed Sahel corresponds to simulated low-albedo biases. Vegetation impacts on albedo are** particularly important at high latitudes where there is snow cover, for example the positive albedo bias in Eastern Siberia is because JULES simulates too few trees and too much grass there. This positive bias in grass cover, affects the snow cover in these regions, which in turn affects the albedo. JULES represents the bending and partial burying of vegetation by snow (Ménard et al., 2014), however, the settings controlling this interaction described in (Sellar et al., 2019) have been tuned for the coupled UKESM1 model rather than the standalone JULES model.

We will reword 'In the observations, shrub and grasses dominate eastern Siberian Taiga in the model instead of tree cover' to read:

'Shrub and grasses dominate eastern Siberian Taiga in the model instead of observed high tree cover (Figure 4)'

Will make this clearer in the text

L247: Perhaps worth mentioning here again that fire was only turned on in a subset of simulations, and not in the official ISIMIP submissions?

Thanks, agreed will add the following to the start of section 3.4:

> In addition to the simulations without fire submitted to the archive, we performed additional simulations with fire and fire feedbacks switched on. These fire simulations provide burnt area and alter vegetation cover, carbon, fluxes, albedo and runoff (Burton et al., 2019).

We have replaced this sentence with:

> We also show to low burnt area in Australia in a common problem across fire models (Hantson et al., 2020a), even in model optimised to burnt area observations (Kelley et al., 2019; Bistinas et al., 2014). This may be due to the unique fire ecology in Northern Australia (Kelley, 2014) and to high uncertainty in observations of burnt area (Giglio et al., 2010). High

burning in South America occurs in areas where cropland fragmentation reduces burnt area beyond the extent of agricultural areas (Kelley et al., 2019a; Andela et al., 2017), which is hard to reproduce in tile and PFT-based models (Hantson et al., 2016). We also simulate too low burnt area in Eurasia. Some of this observed burnt area at these high latitudes comes from peatland fires, which, like most other fire models (Rabin et al., 2017), are not simulated in INFERNO..

**L263: spatial distribution of what?**

The sentence will be modified to clarify that it is the spatial distribution of trees that is being discussed.

**L267: GPPs -> GPP's**

Corrected

**L268: would be useful to know compared to what observed value?**

We compare against estimates from the Global Carbon Budget (Friedlingstein et al., 2020) and will add this reference to the text.

**L270: models' -> model's**

Corrected

**L274: Can you briefly state what these well-documented biases are?**

In most fire models, fire seasons are too long in savannah ecosystems, while timing of the fire season is often too late in human dominated systems (Hanston et al. 2020). These biases have been documented in previous INFERNO evaluations (Burton et al., 2019, 2022, sumbitted; Hantson et al., 2020).We will add this information to the text:

> This is in line with well-documented biases in fire seasonal cycles across all global fire models**, which tend to have longer than observed fire seasons in tropical savannas, and, in human dominated fire regimes, season timing shifted and often late compared to observations** (Hantson et al., 2016, 2020b)**.** Previous evaluations of JULES configurations incorporating INFERNO **also show these biases** (Burton et al., 2019, 2022, submitted; Hantson et al., 2020).

**L284: do you mean the seasonal cycle?**

Corrected

Corrected

Corrected

L305: a bit of a confusing statement (number of trees may lead to higher or lower albedo)

This sentence is highlighting that the albedo is closely related to the landcover, so errors in the simulated landcover compared to observations will result in a bias. Usually too many trees compared to observations will mean that there is a negative albedo bias in the model in terms of tree cover because simulating too many trees mean the land surface is too dark. Too few trees simulated in JULES will mean that there is a positive albedo bias in the model because the land surface is not as dark as it is in the observations. We will include this explanation in the revisied m/s:

> The albedo and land cover area bias are also closely related**. For example, if JULES simulates a larger number of trees than observed, this may lead to a lower albedo than observed or if JULES simulates a lower number of trees this may lead to a higher albedo than observed. Conversely, higher grass cover just North of the observed Sahel corresponds to simulated low-albedo biases**. Vegetation impacts on albedo are particularly important at high latitudes where there is snow cover, for example the positive albedo bias in Eastern Siberia is because JULES simulates too few trees and too much grass there. **Grasses are more readily buried below snow than trees making these areas more reflective** (Sellar et al., 2019)**, which in turn affects the albedo.**

Corrected

The distribution of tropical forests, forest/savanna and North American tree cover, and fire impacts on vegetation and carbon fluxes which we wil clarify in the text

L332: even for hydrological applications with Q simulations degraded?

We would still recommend fire because it is an important process for runoff, even if it degrades performance. We will change the text to make this clear:

> **While fire mostly improves model performance, it does degrade certain vegetation distributions, including lack of larch forest, and runoff. However, fire has a substantial impact on both ecosystem composition and hydrological processes and should therefore still be included when studying impacts under changing climate and environmental conditions.** Therefore, while documentation of the configuration without fire will be useful for anyone using previously submitted results, we recommend using the configuration with fire in future JULES-ES development.

L560 and L565: I think you provided a different reference elsewhere in the text?

We will update the reference in the text to (Dai, 2021)

L561: why weren't longer observations used where available given the GCMs have their own IAV, making a longer evaluation period preferable?

This is an interesting argument but is only valid under equilibrium climates. While the driving GCMs do maintain their own inter-annual variability and trends, that they capture the right trend in climate, and those trends impact on land surface processes, is an important aspect of our evaluation. Using none-overlapping observational and model output means we are at risk of comparing different periods in these long term trends and unfairly penalising simulations that correctly simulate the impacts of climate change.

Figure3: this would be easier to read if the "observed" legends were directly below those panels. Also check albedo unit ("unknown"). Also best to avoid red-green colour schemes.

We will add the fractional unit and adapt to a green-purple colour scheme.

[Figure]

Corrected

Figure 5: No need to repeat legend 5 times? Also would be helpful if the model maps were shown as biases from obs as in previous figures.

We will remove repeated legends, and show the simulations as

a) Fire CCI

b) GFDL-ESM2M    c) HADGEM2-ES

d) IPSL-CM5A-LR    e) MIROC5

difference maps.

The plot shows ensemble ranges. There is a very narrow transition between different shades, which shows that the ensemble only slightly disagrees. We have updated the caption accordingly:

> Figure **1**: Modelled vegetation cover without fire (left) and with fire (middle) compared to observations from ESA CCI Land Cover and Fire (right). Dark green, light green, light brown, dark brown indicates tree, shrub, grass and unvegetated fraction of the latitude band respectively. Shaded transition between colours indicates ensemble range, which is quite narrow indicating agreement across ensemble members. Black hashing indicates burnt area,

with observations taken from MODIS CCI v5.1 (Chuvieco et al., 2018). In **"Model with fire"**, burnt area from the 4 driving models is shown by hatching at 4 different angles.

**Reviewer 2**

**General comment:**

The paper "Description and Evaluation of the JULES-ES set-up for ISIMIP2b" describe the set-up and performance of the Jules-ES for the ISMIP2b protocol very well. The paper describes in concise steps the set-up of the Jules-ES for the ISIMIP2 simulation protocol and presents the comparison to selected observation (runoff, GPP, ET, albedo) for present day comparison very well.

Improvements in the description would be helpful (e.g. how are LU changes tracked within Jules-ES, more info about crop and pasture handling, see specific comments below).

It does fall short on the premise to "describes and evaluates a new modelling methodology to quantify the impacts of climate change on water, biomes and the carbon cycle" and "We simulate a historical and two future scenarios; a mitigation scenario RCP2.6 and RCP6.0, which has very little mitigation. " The paper currently only compares present day conditions to observations, the behavior of Jules-ES for the two future scenarios (impacted by the two contrasting climate change scenarios) are not presented. It would help the reader to judge the performance of Jules-ES in capturing climate change induced changes, if the authors would add a section about the future behavior of the Jules-ES for the contrasting climate change scenarios (e.g. does Jules-ES show an expansion of forest northward in a warmer climate scenario (RCP6.0) compared to a mitigation scenario (RCP2.6)).

I will be happy to review changes to address the above-mentioned concern and the following comments for specific sections.

**Authors response to general comments**

Thank you for your constructive comments we have described how we will address your specific comments in the text below.

Before looking at these more detailed comments, we did want to briefly address the request for future analysis. This paper presents the configuration and evaluation of the modelling framework as the basis for additional studies looking at the critical impacts of climate on terrestrial earth system processes. This is a typical and integral way of conducting climate modelling research and in this respect, papers in GMD have an essential role in meeting this need. Future analysis in the way the reviewer suggests is certainly interesting and important, but it gains the integrity it needs only after this essential documentation and evaluation is complete. Combining evaluation and future impacts in one study blurs the lines between evaluation and policy-relevance sciences. For this reason, it is out of the scope of this paper and the journal.

Documentation and evaluating science through modelling alone resulted in a paper of significant length. All the elements in this current version are important for a proper evaluation, so we have limited the scope to the configuration and evaluation against historical for this paper.

This paper sets the scene for additional studies focusing on future runs and relevant impacts. These later papers will then refer to the evaluation done here without having to redo it, making papers much more manageable. One follow-up paper will include future runs, other ISIMIP ensemble models, and several climate impacts. The future runs are also part of the analysis in the recent UNEP Rapid response report on wildfires (https://wedocs.unep.org/bitstream/handle/20.500.11822/38372/wildfire_RRA.pdf) and in another paper looking at future fires recently submitted by Burton et al. (submitted).

**Responses to specific comments:**

Abstract: "The bias correction reduces the impact of the climate biases present in individual models. We evaluate JULES-ES performance against present-day observations to demonstrate its usefulness for providing required information for impacts such as fire and river flow. "

Very good to see (Figure 1 and S2) that the still quite large difference between GCMS in bias corrected average annual precipitation in the Amazon region is retained very well in the Jules-ES in the modelled catchment scale runoff.

An open question comes to mind, if not bias corrected GCM output would be used to drive Jules-ES, would that result in a poor performance compared to observations or might it show similar results.

The impact of bias correcting the GCM output depends on which Earth system processes you are looking at. Fire for example, is sensitive to the biases in the driving data, with Burton et al. (2022) showing dramatically shifted tropical fire zones when driving a similar JULES-ES configuration with non-bias corrected UK Earth System Model output (Sellar et al., 2019a). However other aspects of the Earth system are less sensitive, and therefore using data that is not bias-corrected does not degrade performance in the same way, such as large-scale vegetation distribution patterns (Sellar et al., 2019).

The aim of bias correcting the data in ISIMIP is to ensure that we can compare the observed and simulated impacts during the historical reference period with a smooth transition into the future period. Bias correcting the data means that we have an accurate starting point, so it is clear when critical thresholds are exceeded in the present and future. We will include the sentences "Bias-correction mean we can compare the observed and simulated impacts during the historical reference period with a smooth transition into the future period" when first introducing the bias-correction methodology, and "Each GCM therefore has a different variability and simulated climate outside of the reference period, but with a smooth transiting going from historical into the reference period, or from the reference period into the future" in the ISIMIP2b protocol description (section 2.3) to make this clear.

Line 38: How more 'computationally efficient' is it, please provide an estimate. e.g. 10% faster or N times faster.

By running JULES stand alone, we can run much faster because we do not have all the feedbacks that we would have in the coupled model. We are running with a prescribed climate which means we do not have to simulate the atmosphere just the land surface. This means we run JULES of the order of 10,000 to 20,000 times faster than UKESM1.

Line 39-41: please clarify, not totally clear what is meant with this.

"while using multi-model climate ensembles sample scientific uncertainty in land surface forcing that would not be possible within a single climate model framework."

In this set-up, we are driving JULES-ES with forcing from four GCMs with different underlying assumptions and parameterisations, which therefore samples different understandings, or uncertainty, in our knowledge of the climate system. If we ran JULES-ES as the land surface scheme of UKESM, then we would only sample one GCM. We have reworded the sentence to reflect this:

An advantage of using JULES-ES as an offline impacts model is that it **uses a prescribed atmosphere, without these feedbacks JULES-ES is more** computationally efficient **than** the closely aligned land surface scheme **coupled to the atmosphere** in UKESM1 (Sellar et al., 2019), **and** using multi-model climate ensembles sample scientific uncertainty in **our understanding of the climate system** that would not be possible within a single climate model framework.

Line 64-66:"tropical broadleaf evergreen trees (BET-Tr)" how is the effect of seasonal drought occurring in various tropical regions handled in the BET-Tr modelling?

Drought impacts all PFTs by reducing photosynthesis via a soil moisture stress function and the impact of humidity deficit on stomatal conductance. Compared to grass PFTs, tree PFTs are less sensitive to humidity and experience different soil moisture stress because of their deeper roots (Harper et al., 2016), but there is no special representation of drought tolerance or phenology for the BET-Tr.

Line 65-66: "C4 grasses, C4 grasses" one of them is a C3, right?

Agreed, thanks for pointing this out this will be modified in the text.

Line 70-72: "How is mineral and manure nitrogen fertilisation handled in the crop modelling? Are there C3 and C4 pasture PFTs? Please provide the crop and pasture PFTs short names (e.g. C3-Cr?)".

The set of grass-like PFTs is natural C3 grasses (C3G), natural C4 grasses (C4G), C3 pasture (C3Pa), C4 pasture (C4Pa), C3 crops (C3Cr) and C4 crops (C4Cr). The only distinction between the natural and pasture grasses, is their geographic distribution – the distribution of pastureland is prescribed as part of the ISIMIP protocol (Frieler et al., 2017), and the model uses the pasture PFTs to ensure only grasses grow on pastureland. Crops have the added distinction of not experiencing nitrogen limitation which requires a fertiliser flux to be diagnosed, no distinction between mineral and manure fertiliser (which we will add to the text) is made and the diagnosed fertiliser flux is entirely taken up by the crop PFTs, having no direct impact on the soil nitrogen stores. Note that the lack of nitrogen limitation increases the productivity of the crop PFTs and increases their litter production, so there will be some indirect increase in soil nitrogen stores and that nitrogen is available to all PFTs.

Line 77: "(urban, ice, lakes)", what about ocean fraction?

We use a land-sea mask to define the fraction of each coastal gridbox that is sea and land, this is fixed throughout the run. We will point out that we consider ocean fraction when determining land area in the revised manuscript.

Line 79-80: "Conversely, when crop and pasture areas are reduced, the natural PFTs are allowed to recolonise the vacated grid box fraction. " Does 'recolonise' entail that new vegetation individuals

establish in the abandoned fractions within the grid box or is the potentially existing Natural fraction just increased?

We will add "The vacated gridbox fraction is initially bare soil and the existing natural PFTs gradually expand their coverage into this area, the rate of expansion is determined by the TRIFFID vegetation dynamics scheme and will follow a succession of faster growing grass PFTs, followed by shrubs and then trees."

Please describe whether Jules-ES performs net or gross LUC transitions.

JULES-ES performs net LUC transitions and will add "net" to the description.

The mentioned "After accounting for land-use, the fractional coverage and biomass of each PFT within a grid box is determined by the TRIFFID dynamic vegetation model. " is not clear, whether the landuse history might be tracked within Jules-ES. Are the various LUC induced fractional areas within the grid box tracked separately within the Jules-ES or the existing landuse fractions just adjusted in size (increases and decreases)?

JULES-ES does not track land-use history, in each grid box each PFT has a single set of state variables, including the prognostic variables: area, height and phenological state.

Line 81-82: "How is grazed or managed pasture grass handled? Any pasture harvest or grazing estimated (e.g., a fraction of grass biomass removed and a portion of the N within the removed biomass returned to the pasture)?" Line 88-90:"Inputs to the land surface are via biological fixation, fertilisation and nitrogen deposition, with losses from the land surface occurring via leaching and gas loss, with Nitrogen deposition being externally provided to the model."  How is N fertilisation forcing handled?

There is no representation of grazing or managed pasture, the pasture PFTs exist to enable the model to force grassy PFTs to grow in the prescribed pastureland area and allow us to output variables specific to pastureland. Older versions of the model used a single pair of C3 and C4 grass PFTs to represent natural grasslands, pasture and croplands, the addition of the crop and pasture PFTs is designed to make it possible to apply crop and pasture specific parameter sets and management activities, but this has not yet been done. No external fertiliser forcing is used, for more information on the representation of fertiliser see the answer to " Line 70-72" above.

Line 90-91:"JULES simulates a nitrogen-limited ecosystem by reducing the net primary productivity if there is insufficient available N to satisfy plant N demand. Any excess carbon is added to the plant respiration. "  N limitation would lead to reduced assimilation, not directly reducing NPP. What excess carbon? Please clarify.

The nitrogen scheme of JULES does not affect GPP but acts through the plant carbon use efficiency. In practice due to the time stepping on the model an update is made to plant respiration. We will update the text to clarify.

Line 100-101: "prescribed population density from HYDE3.2 (Goldewijk et al., 2017) gives human ignitions, and prescribed lightning from LIS/OTD version 2.3.2015 (Cecil, 2006) gives natural ignitions." Those seem to be historic data, what about future scenarios? Is present day population and lightning just recycled into the future?

Population density is from HYDE3.2 for historic and SSP2 for post-2006, as per ISIMIP2b protocol in section 3.2. In other studies, we use lightning climatology per most historic future fire simulations (Rabin et al., 2017; Hantson et al., 2020b). Previous evaluation with JULES-INFERNO, which were compared against observational constraints, shows that burnt area is relatively insensitive to changes in lightning (Mangeon et al., 2016; Burton et al., 2022).

We have moved specific information on driving data to section 3.2 and specified historic and future driving data.

Line 105: "C3 and C4 crop PFTs ", what about pasture PFTs handling in INFERNO?

We parameterise pasture burning as per Burton et al. (2019) as per the previous sentence. Bistinas et al. (2014)  and  Kelley et al. (2019) report increased levels of burning in pasture areas, and this is reflected in Burton et al.'s (2019) parameterisation, and therefore not updated in Burton et al. (2020). We will include a this sentence in the revised manuscript: "[Pasture] C3Pa and C4Pa burning remain as per Burton et al. (2019), which reflects observational constraints that show an increase in burnt area in pasture areas."

Line 117: "model timestep of 1 hour" In supplement the disaggregator was shown with 3-hour timestep. Please clarify that time step discrepancy.

The disaggregator converts to 3 hourly, which JULES then interpolates to 1hr. We shall clarify this in the m/s:

> In the ISIMIP experimental setup, we use the internal disaggregator (Williams and Clark, 2014a) to calculate driving data values at the model timestep of 1 hour. **The method uses the IMOGEN model disaggregator (Huntingford et al., 2010) to initially disaggregate to 3-hourly, which the model linearly interpolates to 1 hour.**

Line 122: "6h for convective rainfall, 1h for large-scale rainfall" How is convective and large-scale rainfall distinguished?

We will add "Convective rainfall occurs when temperature exceeds 288.15°K" to the m/s and point out that "Then the model assumes convective rainfall is more intense and so leads to more runoff and less infiltration into the soil."

Line 129 "See Figure S1 for plots showing that using the disaggregator has little effect on vegetation that we would expect to be 130 influenced by rainfall."  Figure S1 only shows the BET-Tr tree fraction, which is not a sufficient parameter to judge the effect of the time disaggregation. What about biomass, NPP, ET, soil C? Please add the plots and map differences for those important parameters (at least add the plots for your chosen parameters GPP, ET, albedo).

ISIMIP has huge data requirements, and once we finalised development, we only retained the data required for ISMIP submission and evaluating and using the final configuration. We, therefore, did not retain outputs for the variables listed by the reviewer. The extensive evaluation throughout the rest of the study demonstrates that the ISMIP set-up, including the disaggregator, does not degrade model performance. We now realise that picking out the disaggregator for additional supplementary

consideration is a distraction from the paper's main aim. We will remove this plot from the revised manuscript.

Line 136: "(1860-2006) and the RCP2.6 and RCP6.0 future concentration pathways (2006-2099)" The switch between historical and future was 2005 in CMIP5, please clarify?

We will correct this so that the historical in 1860-2005 and RCPs are post-2005.

Line 170-171:"JULES-ES in the form of the suite u-cc669 available via the Met Office Science Repository Service (MOSRS - https://code.metoffice.gov.uk/trac/roses-u see data availability section for information). " MORS requires registration, what about the GMD code and data policy (https://www.geoscientific-model-development.net/policies/code_and_data_policy.html#item4) and its suggested archive of the source code on Zenodo?

JULES is covered by UM licencing so unfortunately; we are unable to share this publicly. A user can register to access the repository should they wish to, but if they don't the Met Office have an agreement in place with GMD that we can share the code with GMD for the purposes of reviewing the paper. There is a process we need to follow which will provide access for you; the necessary information will be added to the manuscript.

Line 185-188:"By assuming there are no losses from the river, we calculated the long-term mean, basin averaged runoff by dividing the river flow at the river mouth by the basin area." How good is that 'no losses' assumption, considering that in many regions river water is used for irrigation, and industrial and household purposes (https://www.globalagriculture.org/report-topics/water.html).

The biggest loss is likely due to irrigation, and much of the other usage e.g. household ends up back in the system on slightly longer timescales via drainage (see e.g. http://www.fao.org/nr/water/aquastat/main/index.stm). We have added that irrigation as a missing process has a substantial impact on arid and semi-arid biomes later in the results (see response to reviewer one comment L193)

We also discuss water extraction and management in the discussion:

> The configurations of JULES can capture the seasonal cycle of many of the largest rivers, although high latitude rivers and managed rivers are generally not captured as well. Including irrigation and structural hydrological developments, such as dams and reservoirs, would likely improve the simulations of managed rivers

**Supplement**

Figure S1: Please add a difference map (e.g. b-a). Why would tree fraction be a good estimator to show the climate disaggregation influence? What about other variables biomass, GPP (or NPP), ET, albedo?

See comment above.

Technical corrections:

Table S2 caption typos:

"global runoff verus Dai" correct: versus.

Thanks this will be corrected

"scores indict whether" correct: indicate

Thanks this will be corrected

Figure S4 please correct the clipping of the ET images on in all 4 panels. The 'Observed albedo' z-scale values overlap and are not readable in the 4 panels.

We will split Figure S4 into supplementary figures for each variable, increasing plots size and improving readability (see response to reviewer 1)

**References**

[revised manuscript text omitted]

---

## Editor Decision (ED1)

The authors have done a good job of improving the text to address the reviewers' comments, and I agree about keeping this paper focused on the historical evaluation. Below, I have a number of corrections that should be made and suggestions that should be considered, but I consider them minor in scope.

Minor comments

- This is silly, but: I really like the color of your tracked changes!
- Thanks for defining ISIMIP in the abstract, but Project should also be capitalized.
- L40–3:
    - Consider adding the "10k-20k more efficient" number you mentioned in your reply to Reviewer 2—that's huge!
    - L41: Comma should be a semicolon
- L73–9:
    - L74: Comma needed after "crops"
    - Please add text explaining that pasture PFTs are not grazed or otherwise managed
- L81: Consider reverting "C3G" and "C4G" to "C3 grasses" and "C4 grasses" for legibility. These abbreviations aren't used elsewhere in the manuscript. (I agree with your revision to include the abbreviations next to the initial PFT definitions, as this will likely help future users of your outputs.)
- L87: Comma should be a semicolon.
- L99:
    - "carbon assimilation" should be "net primary productivity", no? "Carbon assimilation" refers to GPP.
    - What is "nutrient support"?
- L113–9:
    - For readability, please replace "C3Cr and C4Cr PFTs" with "cropland" and "C3Pa and C4Pa" with "pasture".
    - Please briefly summarize how crop and pasture burning differ from natural grasslands.
- L137: "Convective occurs" needs a noun in between—probably "rainfall".
- L171–2:
    - "transiting" should be "transition"
    - "historical" should be "the historical period"
- Table 1 is missing sea-level air pressure. Is this because it's the only one JULES doesn't use? If so, indicate that only used variables are included.
- L195: Comma should be a semicolon.
- L197: Delete Rose, as it's a technical term not previously defined.
- L198: Capitalize Github.
- L208: Sentence should start with "The"; line should end with a comma.
- L209: Missing word? An "ancillary" what?
- L224: Missing word after "smaller".

- L228–9: Are "river channel evaporation and transmission losses" anthropogenic? If so, please add a brief explanation; otherwise, please rework this text.
- L291:
  - "to low" should be "too-low"
  - Replace "Australia in" with "Australia, which is"
  - "model" should be "models"
- L296: Hyphen needed in "too low".
- L297: Replace "like" with "as in".
- L298: "to high" should be "too-high".
- L317: Start sentence with "Without fire," to contextualize "already".
- L357: Comma needed before "or". Also: Consider replacing "or" and following with "and vice versa".
- L358:
  - "North" shouldn't be capitalized.
  - "The observed Sahel"? Delete "observed"?
- L360: ", for example" should be "; for example,"
- L361: Comma needed after "trees".
- L363: ", however" should be "; however,"
- L386: "including lack of larch forest" isn't really complete; please rework. E.g., "for example by simulating too little larch forest". This would also make the sentence easier to read if it were in parentheses rather than being offset with commas.
- L560–1: I'm confused by this sentence, I think because of ", however,".
- Fig. 3:
  - Legend text is much too small. Text in figures should not be much smaller than caption text.
  - Albedo units should be "unitless", not "unknown".
  - Left-most color scheme is improved, but there's still both red and green. Please improve colorblind-friendliness, perhaps by using the same color scheme as the right-most column. ColorBrewer is a good resource, as is the Color Blindness Simulator.
  - Observed albedo color bar: It looks like everything from 0.5 to >0.8 is the same color. You can thus set the "over" triangle to start at 0.5. (Similar issue with observed ET color bar.)
- Fig. 4: Legend text is much too small. Maps are also really small; consider splitting into two figures (perhaps with one in Supplement.)
- Fig. 6:
  - Legend text and axis tick labels are slightly too small.
  - Figure labels are hard to read; please move them off the plots and into the surrounding whitespace.
- Code availability: Please explain that accessing the JULES-ES source code requires registration and that this can be requested at https://jules-lsm.github.io/access_req/JULES_access.html. (This is linked in the Wiltshire et al. paper you referenced, but it's important enough that it should also be here.)

- Fig. S2: Consider converting m/m2 to mm (multiply by 1000), as this is a unit more readers will be familiar with.

- Fig. S3: Caption needs a period at the end.

- Figs. S4 and S5: Legends are still too small to read, as are lat/lon labels. Row/column labels are better but still a bit small.

- In your response to Reviewer 2's comment about Fig. S1, you say "See comment above." What comment, exactly?

---

## Author Response (AR2)

**Response to Reviewers**

Many thanks for your comments, we have addressed them below. Reviewer comments are in blue and our responses are in black.

- This is silly, but: I really like the color of your tracked changes!

  Thanks!

- Thanks for defining ISIMIP in the abstract, but Project should also be capitalized.

  Thank you this has been corrected in the manuscript.

- Consider adding the "10k-20k more efficient" number you mentioned in your reply to Reviewer 2 —that's huge!

  Yeah 😉. Added.

- L41: Comma should be a semicolon

  Thank you this has been corrected in the manuscript.

- L74: Comma needed after "crops"

  Thank you this has been corrected in the manuscript.

- Please add text explaining that pasture PFTs are not grazed or otherwise managed.

  Added text to say this on line 77:

  *Both crop and pasture surface types undergo land-use change according to externally forced time-varying land use, but the pasture is not grazed and is otherwise unmanaged.*

- L81: Consider reverting "C3G" and "C4G" to "C3 grasses" and "C4 grasses" for legibility. These abbreviations aren't used elsewhere in the manuscript. (I agree with your revision to include the abbreviations next to the initial PFT definitions, as this will likely help future users of your outputs.)

  Reverted to using C3 Grasses and C4 Grasses in main text and only using C3G and C4G in the plots.

- L87: Comma should be a semicolon.

  Thank you this has been corrected in the manuscript.

- L99: "carbon assimilation" should be "net primary productivity", no? "Carbon assimilation" refers to GPP. What is "nutrient support"?

  We have change this sentence to read *"This results in a reduced carbon assimilation Net Primary Production (NPP)  when we include nitrogen limitation."*

- L113-119: For readability, please replace "C3Cr and C4Cr PFTs" with "cropland" and "C3Pa and C4Pa" with "pasture".

Reverted to using C3 Crop / C4 Crop and C3 Pasture / C4 Pasture where needed in manuscript.

- L119: Please briefly summarize how crop and pasture burning differ from natural grasslands.
- L137: "Convective occurs" needs a noun in between—probably "rainfall".
  Thanks this has been corrected in the manuscript

- L171–2: "transiting" should be "transition" "historical" should be "the historical period"
  Thanks this has been corrected in the manuscript

- Table 1 is missing sea-level air pressure. Is this because it's the only one JULES doesn't use? If so, indicate that only used variables are included.

  Thanks this was an oversight in reproducing the table and has been added to Table 1 now.

- L195: Comma should be a semicolon.
  Thanks this has been corrected in the manuscript

- L197: Delete Rose, as it's a technical term not previously defined.
  Reference to Rose has now been removed in the manuscript

- L198: Capitalize Github.
  Thanks this has been corrected in the manuscript

- L208: Sentence should start with "The"; line should end with a comma.
  Thanks this has been corrected in the manuscript

- L209: Missing word? An "ancillary" what?
  Thanks this has been corrected in the manuscript

- L224: Missing word after "smaller".
  Thanks this has been corrected in the manuscript, so the sentence now reads:
  *Strong variations between the simulations are also seen in the Brahmaputra basin with some smaller variations in the Chang Jiang basin.*

- L228–9: Are "river channel evaporation and transmission losses" anthropogenic? If so, please add a brief explanation; otherwise, please rework this text.
  Not necessarily, though we realise the sentence makes it sound that way. We have reordered the sentence to read *"In arid and semi-arid basins river flow and runoff tends to be over-estimated, which could be due to missing processes such as river channel evaporation and transmission losses (Haddeland et al., 2011; Döll and Siebert, 2002) and anthropogenic water extraction, primarily irrigation (Richey et al., 2015)"* .

- L291: "to low" should be "too-low" Replace "Australia in" with "Australia, which is" "model" should be "models"
  Thanks this has been corrected in the manuscript

- L296: Hyphen needed in "too low".
  Thanks this has been corrected in the manuscript

- L297: Replace "like" with "as in".

Thanks this has been corrected in the manuscript

- L298: "to high" should be "too-high".

Thanks this has been corrected in the manuscript

- L317: Start sentence with "Without fire," to contextualize "already".

Thanks this has been corrected in the manuscript

- L357: Comma needed before "or". Also: Consider replacing "or" and following with "and vice versa".

Thanks this has been corrected in the manuscript

- L358: "North" shouldn't be capitalized.

Thanks this has been corrected in the manuscript

- "The observed Sahel"? Delete "observed"?

Thanks this has been corrected in the manuscript

- L360: ", for example" should be "; for example,"

Thanks this has been corrected in the manuscript

- L361: Comma needed after "trees".

Thanks this has been corrected in the manuscript

- L363: ", however" should be "; however,"

Thanks this has been corrected in the manuscript

- L386: "including lack of larch forest" isn't really complete; please rework. E.g., "for example by simulating too little larch forest". This would also make the sentence easier to read if it were in parentheses rather than being offset with commas.

Thanks this has been corrected in the manuscript

- L560–1: I'm confused by this sentence, I think because of ", however,".

I think this is in reference to Figure 1, as the caption is on line 650 which is similar to 560?

On this basis, we have amended the text in the Figure 1 caption to be as follows:

**Figure 1: Multi-year mean bias of catchment scale runoff simulated by JULES driven by 4 sets of climate driving data compared to runoff derived from** (Dai, 2021)**. Number of years of observations contributing to the multi-year mean varies depending on catchment and the observations that are available. Observations used are within the period 1980-2006. ISIMIP2b forcing data derived from 4 CMIP5 GCMs: GFDL-ESM2M; HadGEM2-ES; IPSL-CM5A-LR; MIROC5.**

- Fig. 3: Legend text is much too small. Text in figures should not be much smaller than caption text. Albedo units should be "unitless", not "unknown". Left-most color scheme is improved, but there's still both red and green. Please improve colorblind-friendliness, perhaps by using the same color scheme as the right-most column. ColorBrewer is a good resource, as is the Color Blindness Simulator. Observed albedo color bar: It looks like everything from 0.5 to >0.8 is the

same color. You can thus set the "over" triangle to start at 0.5. (Similar issue with observed ET color bar.)

We have made changes to the figure in the manuscript, as shown below. I think we have addressed all issues:

- Font sizes increased everywhere
- Legends clearer, and no redundancy
- Colour schemes are now colour-blind safe
- Albedo units corrected

[Figure]

Fig. 4: Legend text is much too small. Maps are also really small; consider splitting into two figures (perhaps with one in Supplement.)

We have made changes to the figure. As suggested, we reduced the complexity of the figure by calculating an ensemble mean vegetation distribution, and just showing the anomaly of that in comparison to observations. There were very minimal differences between each ensemble member. The figure now looks like this in the manuscript:

[Figure]

ISIMIP: PFT Fractions (Fire Off)

- Fig. 6: Legend text and axis tick labels are slightly too small. Figure labels are hard to read; please move them off the plots and into the surrounding whitespace.
  We have made changes to the figure, which now looks like this in the manuscript:

[Figure]

- Code availability: Please explain that accessing the JULES-ES source code requires registration and that this can be requested at https://jules-lsm.github.io/access_req/JULES_access.html. (This is linked in the Wiltshire et al. paper you referenced, but it's important enough that it should also be here.)

  Added text as in Wiltshire et al to explain access to JULES code:

  *Note that to view and use the JULES-ES source code, access will be required to the Met Office Science Repository Service (https://code.metoffice.gov.uk/trac/home) and is available to those who have signed the JULES user agreement. The easiest way to access the repository is by completing the online form to register here: http://jules-lsm.github.io/access_req/JULES_access.html*

- Fig. S2: Consider converting m/m2 to mm (multiply by 1000), as this is a unit more readers will be familiar with.
  We have made this change

- Fig. S3: Caption needs a period at the end.
  Added

- Figs. S4 and S5: Legends are still too small to read, as are lat/lon labels. Row/column labels are better but still a bit small.

  Done. Figures changed to increase font sizes in legends, lat/lon labels and row/column labels. As follows:

ISIMIP: Seasonal Mean ET

[Figure]

[Figure]

- In your response to Reviewer 2's comment about Fig. S1, you say "See comment above." What comment, exactly?

It is in response to the reviewers comment about line 129 of the original m/s, were we wrote:

ISIMIP has huge data requirements, and once we finalised development, we only retained the data required for ISMIP submission and evaluating and using the final configuration. We, therefore, did not retain outputs for the variables listed by the reviewer. The extensive evaluation throughout the rest of the study demonstrates that the ISMIP set-up, including the disaggregator, does not degrade model performance. We now realise that picking out the disaggregator for additional supplementary consideration is a distraction from the paper's main aim. We will remove this plot from the revised manuscript.

---

## Author Response (AR3)

**Response to Reviewers**

Many thanks for your comments, we apologise for missing these points in your original review. We address these as follows, with the comments in blue and our comments in black.

L119-20: Please briefly summarize how crop and pasture burning differ from natural grasslands; don't just refer to Burton et al. (2019). (I think you missed this in my original review.)

Apologies, we did miss it. Pasture burns at the same rate as the natural grass PFT, and we have adapted this sentence to read:

C3 Pasture and C4 Pasture burn at the same rate as natural grasses as per Burton et al. (2019), which reflects observational constraints that show an increase in burnt area in pasture areas (Kelley et al., 2019; Bistinas et al., 2014).

Thanks for the rework of Fig. 4. However, the caption needs to be updated to match.

Many thanks for pointing this out. The caption now reads:

Figure 4: Observed vegetation fractional cover. First column is derived from ESACCI Land Cover (v2.0.7) for 2010 (Harper et al., 2022). The second column is the difference between the observations and the simulated ensemble mean. Top to bottom show tree, shrub, grass and unvegetated (bare) fraction.

While I was looking through your revisions, I also noted the following

- L41: "10-20k" by itself is unclear. Please revise that bit of the sentence to read "without these feedbacks JULES-ES is is 10,000 to 20,000 times more computationally efficient…"

L172-3: "historical period" should be "the historical period"

We have made these changes as suggested